# N-hydroxypipecolic acid triggers systemic acquired resistance through extracellular NAD(P)

Qi Li [1], Mingxi Zhou [1,2], Shweta Chhajed [3], Fahong Yu[4], Sixue Chen[5], Yanping Zhang [4] & Zhonglin Mou [1,2] ✉

Systemic acquired resistance (SAR) is a long-lasting broad-spectrum plant defense mechanism induced in distal systemic tissues by mobile signals generated at the primary infection site. Despite the discoveries of multiple potential mobile signals, how these signals cooperate to trigger downstream SAR signaling is unknown. Here, we show that endogenous extracellular nicotinamide adenine dinucleotide (phosphate) [eNAD(P)] accumulates systemically upon pathogen infection and that both eNAD(P) and the lectin receptor kinase (LecRK), LecRK-VI.2, are required in systemic tissues for the establishment of SAR. Moreover, putative mobile signals, e.g., N-hydroxypipecolic acid (NHP), trigger de novo systemic eNAD(P) accumulation largely through the respiratory burst oxidase homolog RBOHF-produced reactive oxygen species (ROS). Importantly, NHP-induced systemic immunity mainly depends on ROS, eNAD(P), LecRK-VI.2, and BAK1, indicating that NHP induces SAR primarily through the ROS-eNAD(P)-LecRK-VI.2/BAK1 signaling pathway. Our results suggest that mobile signals converge on eNAD(P) in systemic tissues to trigger SAR through LecRK-VI.2.

Both animals and plants have evolved sophisticated immune systems to fend off microbial pathogens. Compared to animals, plants lack mobile innate immune cells and adaptive immunity, but are equipped with large numbers of cell surface pattern recognition receptors (PRRs) and intracellular nucleotide-binding domain leucine-rich repeat receptors (NLRs) to surveil potential microbial pathogens and mount immune responses[1–3]. Upon infection, PRRs recognize conserved microbial features named pathogen-associated molecular patterns (PAMPs), activating PAMP-triggered immunity (PTI). PRRs can also sense host-derived damage-associated molecular patterns (DAMPs), initiating DAMP-triggered immunity (DTI). NLRs detect pathogen effectors or effector-caused cellular perturbation, inducing effector-triggered immunity (ETI). Concomitant with the activation of PTI, DTI, and ETI, mobile signals produced at the primary infection site are transported through the vascular system to distal systemic tissues of the plant, resulting in the establishment of a long-lasting defense mechanism called systemic acquired resistance (SAR) that provides whole-plant resistance against subsequent infections by a broad spectrum of pathogens[4,5].

In the past three decades, several chemically unrelated molecules have been proposed as SAR mobile signals, including the lipid transfer protein DIR1[6], salicylic acid (SA) and its derivative methyl SA[7–9], dehydroabietinal[10], azelaic acid (AzA)[11], glycerol-3-phosphate (G3P)[12], pipecolic acid (Pip) and its derivative N-hydroxy-Pip (NHP)[13–15], extracellular NAD(P) [eNAD(P)][16], monoterpenes[17], and phased small RNAs[18]. Among these signals, DIR1 has been demonstrated to move down the leaf petiole to distant leaves upon SAR induction[19]. Moreover, nitric oxide (NO) and reactive oxygen species (ROS) have been shown to

[1]Department of Microbiology and Cell Science, University of Florida, P.O. Box 110700, Gainesville, FL 32611, USA. [2]Plant Molecular and Cellular Biology Program, University of Florida, P.O. Box 110690, Gainesville, FL 32611, USA. [3]Department of Biology, University of Florida, P.O. Box 118525, Gainesville, FL 32611, USA. [4]Interdisciplinary Center for Biotechnology Research, University of Florida, P.O. Box 103622, Gainesville, FL 32610, USA. [5]Department of Biology, University of Mississippi, Oxford, MS 38677-1848, USA. ✉e-mail: zhlmou@ufl.edu

amplify SAR signals[20,21]. It has been proposed that Pip, NO, ROS, AzA, and G3P function in a linear pathway in parallel with SA-derived signaling in the induction of SAR[9,21]. Despite these discoveries, how mobile signals act together to trigger downstream SAR signaling in systemic tissues remains unknown.

Exogenous NAD(P) has been shown to induce robust local immune responses in *Arabidopsis*[22,23]. We have also shown that prolonged treatment with NAD(P)⁺ induces weak but significant systemic immunity[16], suggesting a role for NAD(P) in SAR. Here, we found that a single infiltration of lower leaves on *Arabidopsis* plants with 1 mM NAD(P)⁺ induced strong systemic immunity against the virulent bacterial pathogen *Pseudomonas syringae* pv. *maculicola* ES4326 (*Psm*) when the challenge-inoculation of upper systemic leaves was conducted between 2 and 8 h after the NAD(P)⁺ treatment (Supplementary Fig. 1a–c). NAD(P)⁺-induced systemic immunity was concentration-dependent, with >0.8 mM NAD⁺- and >0.6 mM NADP⁺-induced resistance being comparable to that triggered by biological induction of SAR (Supplementary Fig. 1d), indicating that exogenous NAD(P)⁺ is a potent SAR inducer.

We have previously reported that exogenously added NAD⁺ moves systemically and that the lectin receptor kinase (LecRK), LecRK-VI.2, is a putative NAD(P) receptor and plays a pivotal role in biological induction of SAR[16]. We have also shown that BRASSINOSTEROID INSENSITIVE1-ASSOCIATED KINASE1 (BAK1) constitutively associates with LecRK-VI.2 and functions in eNAD(P) signaling and SAR[16]. Although these results suggest that eNAD(P) is a potential SAR mobile signal, the function of endogenous eNAD(P) in SAR remains unclear.

Here, we show that endogenous eNAD(P) is absolutely required for SAR induction as the converging point of mobile signals and demonstrate that NHP induces SAR through the ROS-eNAD(P)-LecRK-VI.2/BAK1 signaling pathway in systemic tissues.

## Results

### Endogenous NAD(P) is indispensable for SAR

Since NAD(P) is believed to leak into the extracellular space upon pathogen-caused cell damage[22], we tested whether cellular NAD(P) plays a role in SAR. To this end, we took advantage of the previously reported NAD biosynthesis mutant *flagellin-insensitive4-3* (*fin4-3*) that carries a T-DNA insertion toward the 3′ end of the *FIN4* gene[24]. *FIN4* encodes the chloroplastic enzyme aspartate oxidase that catalyzes the first irreversible step in de novo NAD biosynthetic pathway[24,25]. The *fin4-3* mutant is smaller than wild type and accumulates significantly reduced NAD(P) levels[26] (Fig. 1a–c). A *35S:FIN4* transgene complemented the *fin4* morphology and restored NAD(P) levels in *fin4-3* to the wild-type level (Fig. 1a–c). We then tested biological induction of SAR with *Psm* and the avirulent pathogen *P. syringae* pv. *tomato* DC3000 carrying the effector gene *avrRpt2* (*Pst avrRpt2*) in *fin4-3* and two independent complementation lines. Wild type and an SAR null mutant, *npr1-3*, were included as positive and negative controls, respectively. Surprisingly, the *fin4-3* mutant exhibited severe SAR defects like those observed in *npr1-3*[27]. SAR-mediated resistance to *Psm* and induction of three SAR marker genes, *PR1*, *ALD1*, and *FMO1* were almost completely abolished in *fin4-3* (Fig. 1d–f and Supplementary Fig. 2a, b). Importantly, these SAR defects were fully complemented by the *35S:FIN4* transgene (Fig. 1d–f and Supplementary Fig. 2a, b), indicating that NAD(P) synthesized through the FIN4-mediated de novo pathway plays a crucial role in SAR establishment. To further examine the effect of the *fin4* mutation on SAR signaling, we compared the transcriptomes in the systemic leaves of wild-type and *fin4-3* plants at 48 h after SAR induction and identified a total of 1924 differentially expressed genes (DEGs) (Supplementary Data 1). Gene ontology (GO) analysis of these DEGs revealed that those involved in plant immune responses were significantly enriched (Supplementary Fig. 2c). Specifically, expression of the SA biosynthesis genes *ICS1*, *EDS5*, and *PBS3*, the NHP biosynthesis genes *ALD1*, *SARD4*, and *FMO1*, the G3P

biosynthesis gene *NHO1*, and a group of NPR1 target genes, including *PR1*, *PR2*, and *PR5*, was downregulated in *fin4-3* (Fig. 1g and Supplementary Data 2), confirming the importance of endogenous NAD(P) in SAR.

### NAD(P)⁺ triggers immune responses in the extracellular space

Since the plasma membrane is highly impermeable to NAD(P)[25,28], exogenously added NAD(P) might function in the extracellular space to trigger immune responses. In line with this hypothesis, the reduced total NAD(P) levels in *fin4-3* did not affect the mutant's responses to 0.2 mM NAD⁺ and 0.4 mM NADP⁺ (Fig. 2a, b), which are relatively low concentrations for immune activation[22,23]. As FIN4 catalyzes the first step in the de novo pathway of NAD biosynthesis, we treated *fin4-3* with nicotinic acid (NA), a cell membrane permeable intermediate that can be converted to NAD through the salvage pathway, to restore intracellular NAD level in the mutant[24]. Infiltration of *fin4-3* leaves with 4 mM NA restored NAD levels within 4 h (Supplementary Fig. 3a). We then treated *fin4-3* and wild-type plants with 4 mM NA and 0.2 mM NAD⁺ and monitored total NAD levels at 4 h after the treatment. Of note, total NAD includes both intracellular and extracellular NAD. Consistent with the fact that cellular NAD homeostasis is tightly regulated[25,29], NA did not change the NAD level in wild type but restored that in *fin4-3* (Fig. 2c). As conversion of NA into NAD occurs inside the cell[25], NA restored the intracellular NAD level in *fin4-3*. On the other hand, NAD⁺ increased the total NAD levels in both wild type and *fin4-3*, though the increase in the wild type is not statistically significant (Fig. 2c). Because the plasma membrane is impermeable to NAD⁺[25,28], these increases may be due to the added NAD⁺ in the extracellular space, although breakdown products of NAD⁺ might be taken up by *fin4-3* to synthesize intracellular NAD. Regardless, NAD⁺, but not NA, induced the expression of the defense genes *FRK1*, *ALD1*, and *FMO1* in wild-type and *fin4-3* plants (Fig. 2d), even though NAD⁺ and NA elevated total NAD to similar levels in *fin4-3* (Fig. 2c). Moreover, under our experimental conditions, *fin4-3* exhibited compromised basal resistance to *Psm*[24] (Supplementary Fig. 3b, c). NA treatment restored basal resistance to *Psm* (the resistance activated by *Psm* on the wild-type plants) in *fin4-3*, whereas only NAD⁺ treatment induced resistance to *Psm* in wild type and *fin4-3* (Fig. 2e). Furthermore, NAD⁺ treatment, but not NA treatment, induced systemic immunity in wild type and *fin4-3* (Fig. 2f). Taken together, these results suggest that NAD(P)⁺ is immunogenic when present in the extracellular space.

### The eNAD(P) receptor LecRK-VI.2 is required in systemic tissues

We have previously shown that the lectin receptor kinase (LecRK), LecRK-VI.2, is a potential receptor for extracellular NAD(P) [eNAD(P)][16]. To find out where eNAD(P) initiates SAR signaling, we determined the requirement of its receptor LecRK-VI.2 in local and systemic tissues for SAR establishment. A stringent glucocorticoid-inducible system, pOpON[30], was used to spatially control the expression of the *LecRK-VI.2* transgene in the *lecrk-VI.2* mutant background by dexamethasone (Dex) application. The bidirectional promoter in the pOpON vector allows selection of transgenic lines based on the induction of the β-glucuronidase (GUS) reporter. Two homozygous *Dex:LecRK-VI.2/lecrk-VI.2* transgenic lines that showed no background and strongly induced GUS activity and *LecRK-VI.2* transcription upon Dex application were selected for further investigation (Fig. 3a, b). Of note, GUS activity was not detectable in the upper systemic leaves of *Dex:LecRK-VI.2/lecrk-VI.2* plants after infiltration of lower leaves with Dex (Fig. 3a), indicating that Dex itself does not move systemically. We first tested the effects of local and systemic induction of the expression of *LecRK-VI.2* on NAD(P)⁺-induced systemic immunity in *Dex:LecRK-VI.2/lecrk-VI.2* plants (Supplementary Fig. 4a). NAD(P)⁺ treatment conveyed wild-type levels of systemic immunity in *Dex:LecRK-VI.2/lecrk-VI.2* plants only when *LecRK-VI.2* was induced in the upper systemic leaves (Fig. 3c–f),

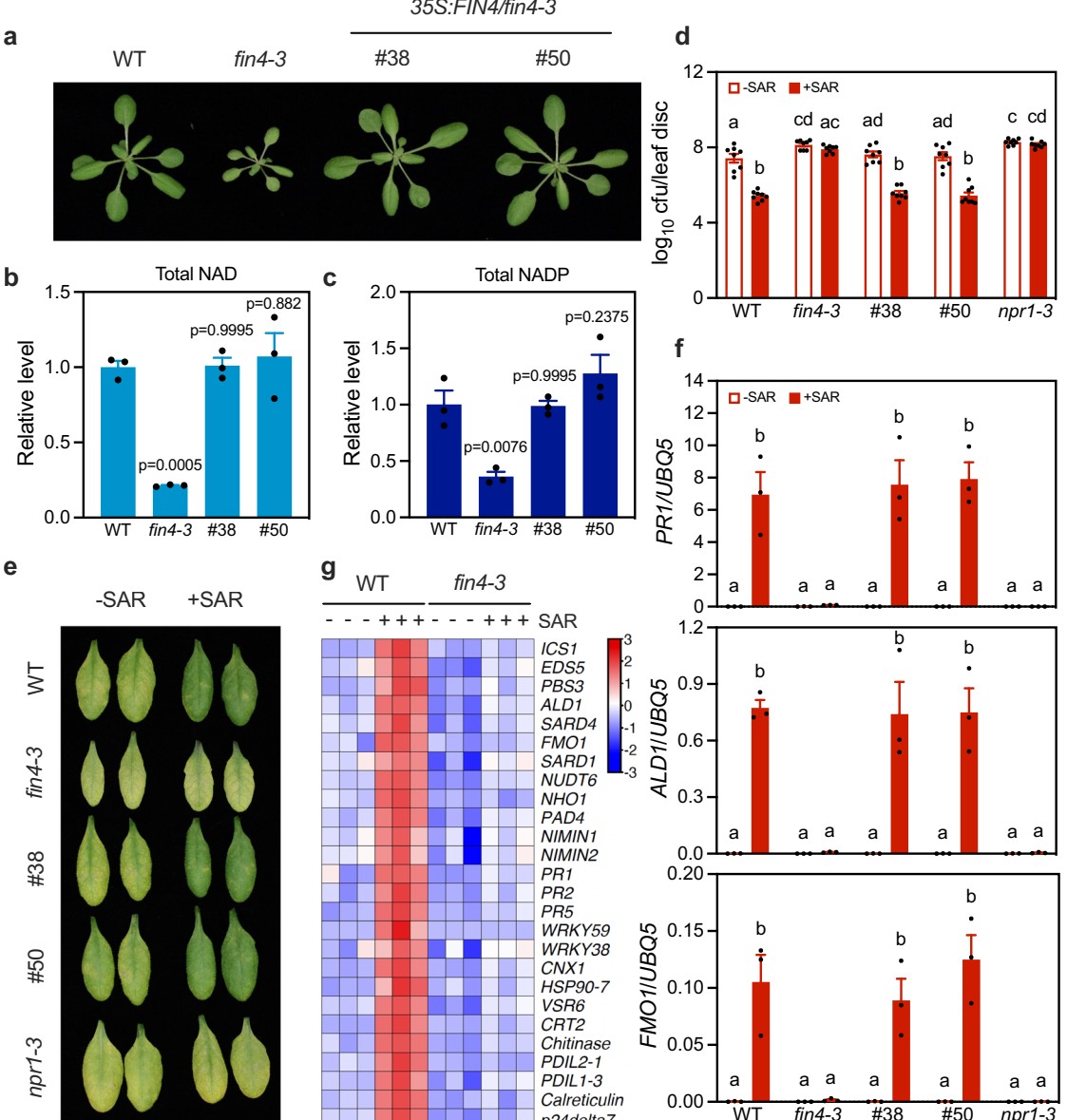

**Fig. 1 | The *fin4-3* mutation compromises biological induction of SAR.**
**a** Morphology of three-week-old wild-type (WT), *fin4-3*, and two *35 S:FIN4/fin4-3* complementation lines. **b, c** Total NAD (**b**) and NADP (**c**) levels in the indicated genotypes. Leaf samples of four-week-old plants were collected. Values are expressed relative to the NAD(P) levels in WT, which were arbitrarily set to 1, allowing comparison across experiments. Bars represent means ± SE (*n* = 3 independent leaf samples). The *35 S:FIN4* transgene restored total NAD(P) levels in *fin4-3* (one-way ANOVA with Tukey's test). The experiment was repeated with similar results. **d** Biological induction of SAR in the indicated genotypes. cfu: colony-forming unit. Bars represent means ± SE (*n* = 8 independent leaf disks). Different letters denote significant differences (one-way ANOVA with Tukey's test; *p* values are shown in the Source Data file). The experiment was conducted three times with

similar results. **e** Disease symptoms on the inoculated systemic leaves at 72 h post-inoculation (hpi) in (**d**). **f** SAR induction-activated systemic expression of *PR1*, *ALD1*, and *FMO1* in the indicated genotypes. Three lower leaves on each plant were infiltrated with 1 mM $MgCl_2$ (-SAR) or *Psm* (+SAR). After 48 h, one systemic leaf on the plant was collected for gene expression assay. Bars represent means ± SE (*n* = 3 independent total RNA samples). Different letters denote significant differences (one-way ANOVA with Tukey's test; *p* values are shown in the Source Data file). The experiment was repeated with similar results. **g** A heatmap of transcript levels of the indicated genes in the systemic leaves of WT and *fin4-3* at 48 h after SAR induction based on the RNA-seq data. The color bar represents normalized $log_2$(mapped reads).

indicating that LecRK-VI.2 is required in systemic leaves for NAD(P)⁺ treatment to induce systemic immunity. We then determined the effects of local and systemic expression of *LecRK-VI.2* on biological induction of SAR in *Dex:LecRK-VI.2/lecrk-VI.2* plants (Supplementary Fig. 4b). As shown in Fig. 3g, h, strong biological SAR was established in *Dex:LecRK-VI.2/lecrk-VI.2* plants only when *LecRK-VI.2* was induced in the upper systemic leaves. These results demonstrate that LecRK-VI.2 is required in systemic leaves for SAR establishment and suggest that eNAD(P) must accumulate in systemic tissues to trigger SAR.

## eNAD(P) accumulated at the primary infection site moves systemically

In line with the requirement of LecRK-VI.2 in systemic tissues, biological induction of SAR triggered significant accumulation of eNAD(P) in systemic leaves (Fig. 4a, b), while total NAD(P) levels in the systemic leaves were not significantly changed (Supplementary Fig. 5a, b). Note that eNAD(P) levels were assessed by analyzing apoplastic washing fluids (AWFs) of the indicated leaves. To compare systemic eNAD(P) accumulation during SAR induction and

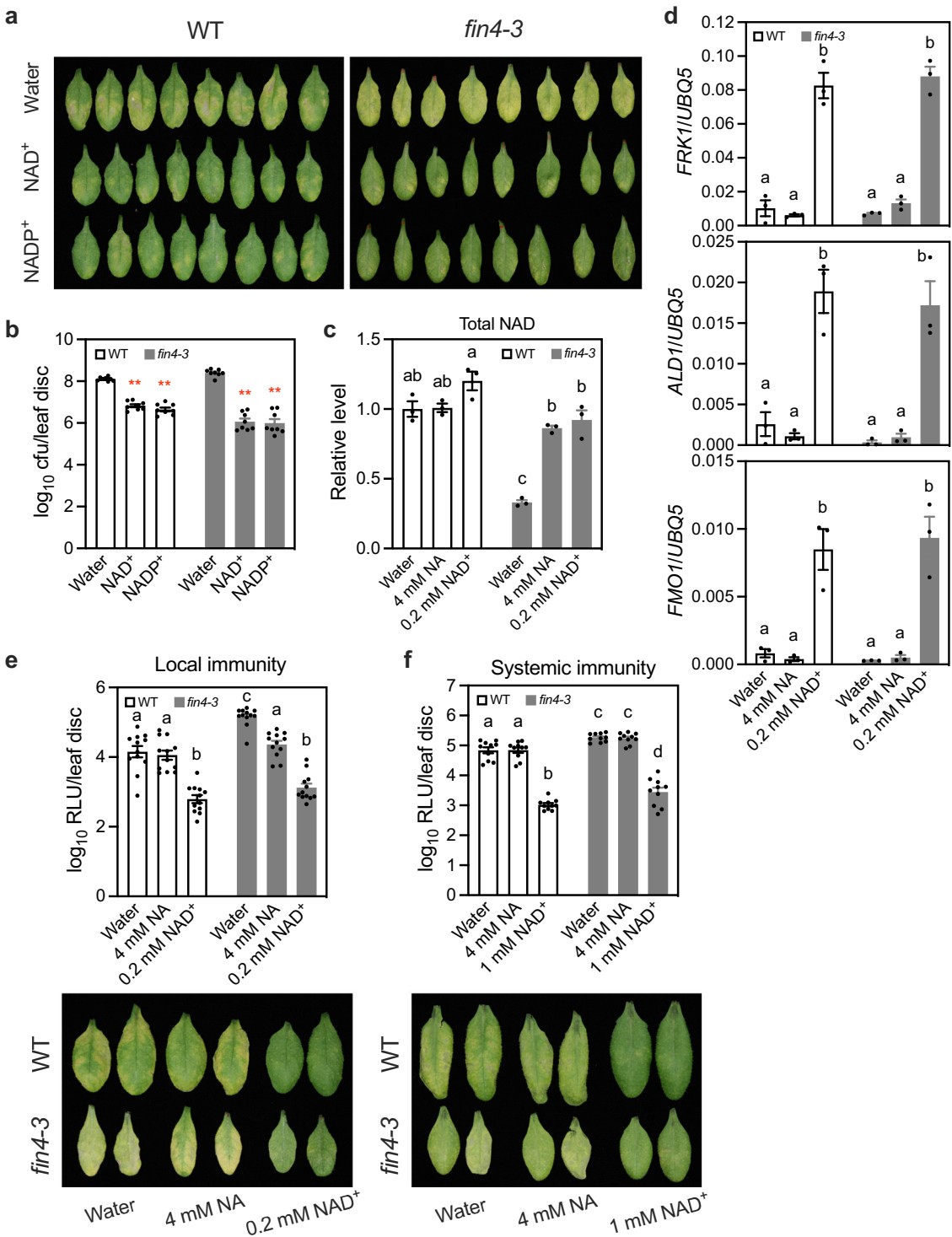

**Fig. 2 | NAD(P)+ is a strong immune activator in the extracellular space.**
**a**, **b** NAD(P)+-induced local immunity in wild type (WT) and *fin4-3*. Two leaves on each plant were infiltrated with 0.2 mM NAD+, 0.4 mM NADP+, or water. Four hr later, the infiltrated leaves were inoculated with *Psm*. Photos in (**a**) and samples in (**b**) were taken 72 hpi. Bars in (**b**) represent means ± SE (*n* = 8 independent leaf disks). Asterisks denote significant differences between NAD(P)+- and water-treated samples (two-tailed *t* test; *p* values are shown in the Source Data file). The experiment was conducted three times with similar results. **c**, **d** Total NAD (including intracellular and extracellular NAD) levels (**c**) and induction of *FRK1*, *ALD1*, and *FMO1* (**d**) in WT and *fin4-3* treated with NA or NAD+. Two leaves on each plant were infiltrated with 4 mM NA, 0.2 mM NAD+ or water. The infiltrated leaves were

collected 4 h later. Bars represent means ± SE (*n* = 3 independent leaf samples (**c**) or total RNA samples (**d**)). Different letters denote significant differences (one-way ANOVA with Tukey's test; *p* values are shown in the Source Data file). The experiments were repeated with similar results. **e**, **f** NA- and NAD+-induced local (**e**) and systemic (**f**) immunity in WT and *fin4-3*. Two (**e**) or three leaves (**f**) on each plant were infiltrated with the indicated concentrations of NA, NAD+ or water. Four hr later, either the infiltrated leaves (**e**) or one upper systemic leaf (**f**) was inoculated with *Psm*. Samples were taken 72 hpi. Bars represent means ± SE (*n* = 8 independent leaf disks). Different letters denote significant differences (one-way ANOVA with Tukey's test; *p* values are shown in the Source Data file). RLU relative light unit. The experiments were performed three times with similar results.

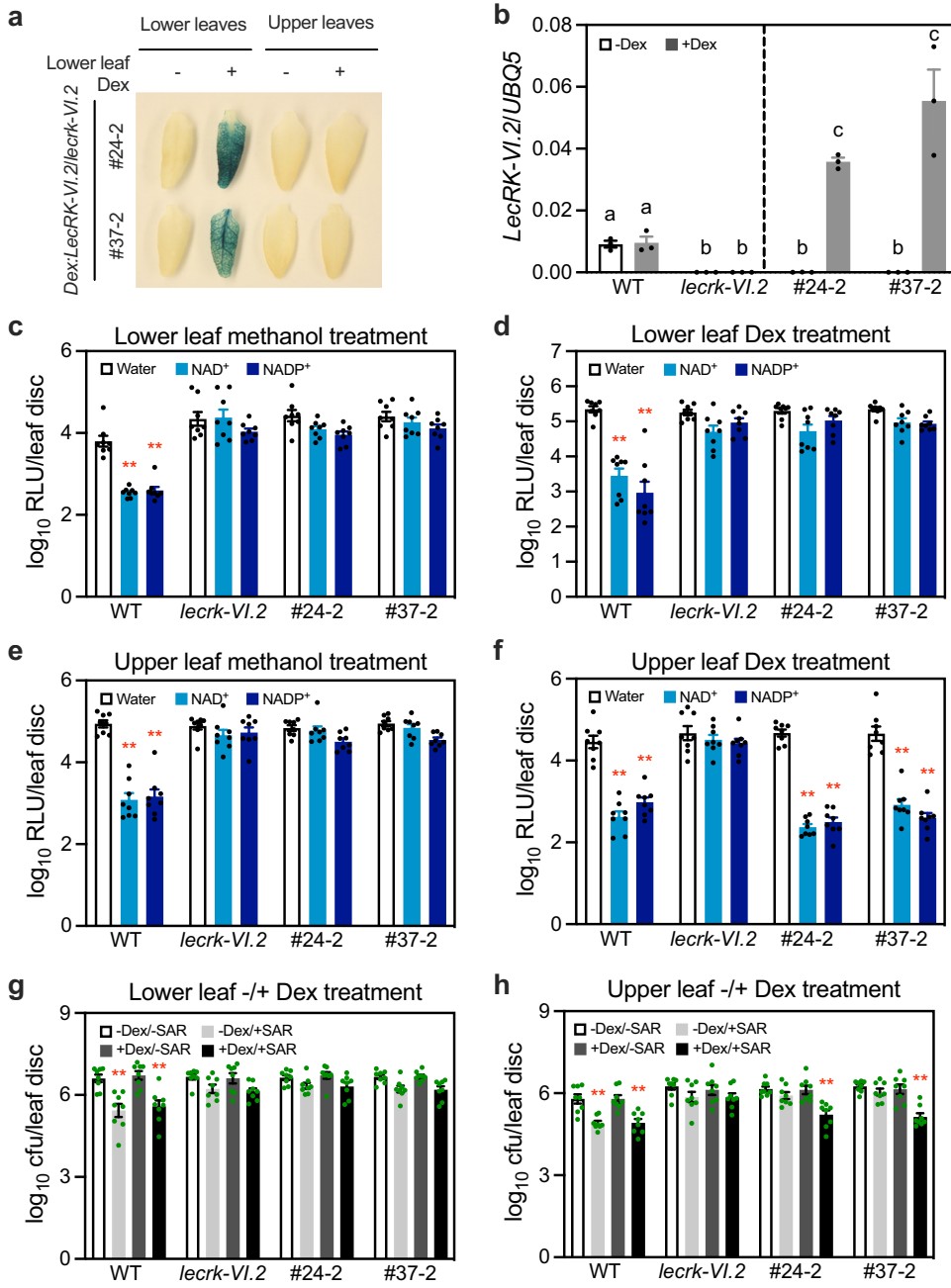

**Fig. 3 | Systemic LecRK-VI.2 is required for the induction of SAR. a** GUS expression in Dex-infiltrated lower leaves and upper noninfiltrated systemic leaves of the *Dex:LecRK-VI.2/lecrk-VI.2* plants. Three lower leaves on each plant were infiltrated with 50 μM Dex or 0.1% methanol (-Dex). Twenty-four hr later, the infiltrated lower leaves and one upper systemic leaf from each plant were subjected to GUS staining. **b** Induction of the *LecRK-VI.2* transgene in the *Dex:LecRK-VI.2/lecrk-VI.2* plants by Dex application. Two leaves on each plant were infiltrated with 50 μM Dex or 0.1% methanol (-Dex). The infiltrated leaves were collected 24 h later. Bars represent means ± SE (*n* = 3 independent total RNA samples). Different letters denote significant differences (one-way ANOVA with Tukey's test; *p* values are shown in the Source Data file). The experiment was repeated with similar results. **c**–**f** NAD(P)+-induced systemic immunity in *Dex:LecRK-VI.2/lecrk-VI.2* plants with lower leaves (**c** and **d**) or upper systemic leaves (**e** and **f**) being pretreated with or without Dex. Three lower leaves (**c** and **d**) or one upper systemic leaf (**e** and **f**) on

each plant was infiltrated with 50 μM Dex (**d** and **f**) or 0.1% methanol (**c** and **e**). Twenty-four hr later, NAD(P)+-induced systemic immunity was determined. Bars represent means ± SE (*n* = 8 independent leaf disks). Asterisks denote significant and wild-type levels of differences between NAD(P)+- and water-induced samples (two-tailed *t* test; *p* values are shown in the Source Data file). The experiments were performed three times with similar results. **g**, **h** Biological induction of SAR in *Dex:LecRK-VI.2/lecrk-VI.2* plants with lower leaves (**g**) or upper systemic leaves (**h**) being pretreated with or without Dex. Three lower leaves (**g**) or one upper systemic leaf (**h**) on each plant was infiltrated with 50 μM Dex or 0.1% methanol (-Dex). Twenty-four hr later, biological induction of SAR was determined. Bars represent means ± SE (*n* = 8 independent leaf disks). Asterisks denote significant and wild-type levels of differences between -SAR and +SAR samples (two-tailed *t* test; *p* values are shown in the Source Data file). The experiments were performed three times with similar results.

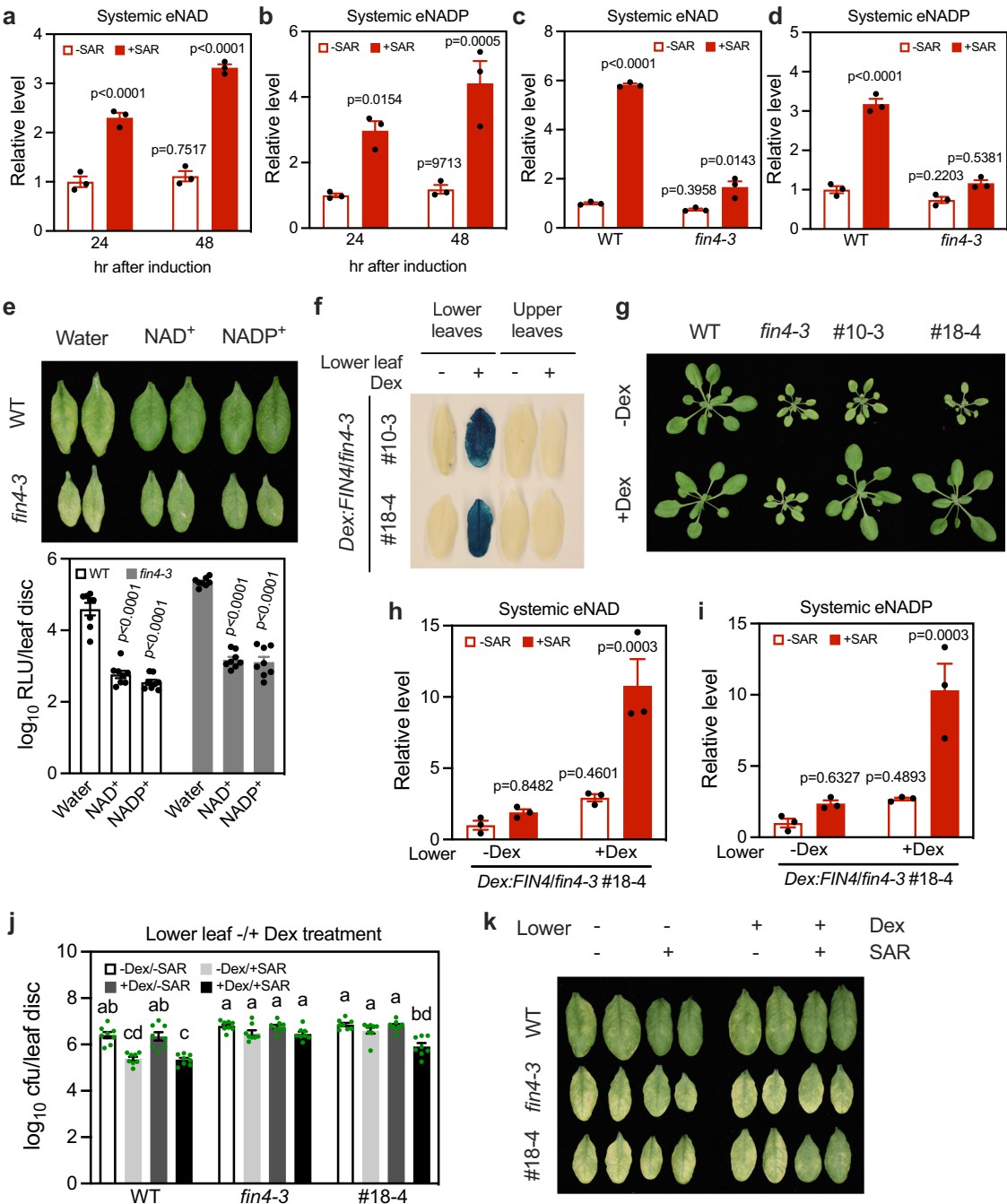

exogenous NAD(P)+ treatment, we infiltrated lower leaves with 1 mM MgCl₂, *Psm*, water, 0.5 mM NAD⁺, or 1 mM NAD⁺. Systemic leaves were collected four hr after the water and NAD⁺ treatments and 48 h after the MgCl₂ and *Psm* treatments for eNAD assays. The systemic eNAD levels upon SAR induction was in the range of 30–40 μg/g fresh weight (FW), which is comparable to those after 0.5 mM NAD⁺ treatment (Supplementary Fig. 6). Furthermore, the *fin4-3* mutant accumulated dramatically reduced eNAD(P) levels in the local leaves inoculated with *Psm* (Supplementary Fig. 7a, b) and SAR induction-triggered systemic accumulation of eNAD(P) was also largely diminished in *fin4-3* compared with that in the wild type (Fig. 4c, d). Importantly, exogenous NAD(P)+ was able to induce similar levels of systemic immunity in the wild-type and *fin4-3* plants (Fig. 4e). These results indicate that the compromised SAR phenotype of *fin4-3* is due to reduced eNAD(P) accumulation in systemic leaves.

We have shown that ³²P from locally added ³²p-NAD⁺ moved systemically[16], suggesting that either NAD⁺ or its metabolites might move systemically to induce systemic immunity in plants. To clarify this finding, we infiltrated ¹³C-NAD⁺ into lower leaves on *Arabidopsis* plants. Both the infiltrated leaves and upper systemic leaves were collected for tandem mass spectrometry analysis of ¹³C-NAD⁺. Whereas unlabeled NAD(P)+ levels were not significantly different in the infiltrated and systemic leaf extracts (Supplementary Fig. 8a–e), the ¹³C-NAD⁺ level detected in the systemic leaf extracts was approximately 9.3% of that in the infiltrated leaf extracts (Supplementary Fig. 8f), which is close to the ratio (-10.8%) obtained with ³²p-NAD⁺[16], indicating that exogenously added ¹³C-NAD⁺ moved systemically in its intact form.

To determine whether endogenous eNAD(P) is mobile, we took advantage of the pOpON system to spatially control the expression of the *FIN4* transgene, thus NAD biosynthesis, in the *fin4-3* mutant

**Fig. 4 | eNAD(P) is a potential mobile SAR signal. a, b** SAR induction-triggered systemic accumulation of eNAD(P). The eNAD(P) levels were assessed by analyzing apoplastic washing fluids (AWFs) of the indicated leaves. Three lower leaves on each wild-type plant were infiltrated with *Psm* (+SAR) or 1 mM MgCl$_2$ (-SAR). One upper systemic leaf on each plant was collected at the indicated times. Values are expressed relative to the eNAD(P) levels in -SAR samples at 24 h, which were arbitrarily set to 1, allowing comparison across experiments. Bars represent means ± SE (*n* = 3 independent AWF samples). SAR induction caused significant eNAD(P) accumulation in the systemic leaves (one-way ANOVA with Tukey's test). The experiment was repeated with similar results. **c, d** SAR induction-triggered systemic accumulation of eNAD(P) in wild type (WT) and *fin4-3* at 48 h after the induction. Values are expressed relative to the eNAD(P) levels in WT/-SAR samples, which were arbitrarily set to 1. Bars represent means ± SE (*n* = 3 independent AWF samples). The SAR induction-induced systemic eNAD(P) accumulation was significantly reduced in *fin4-3* (one-way ANOVA with Tukey's test). The experiment was repeated with similar results. **e** NAD(P)⁺-induced systemic immunity in WT and *fin4-3*. Bars represent means ± SE (*n* = 8 independent leaf disks). NAD(P)⁺ induced similar levels of systemic immunity in the WT and *fin4-3* plants (two-tailed *t* test). The photo was taken 72 hpi. The experiment was conducted three times with similar results. **f** GUS expression in Dex-infiltrated lower leaves and upper noninfiltrated systemic leaves

of *Dex:FIN4/fin4-3* plants. Leaves were collected 24 h after the treatment. **g** Morphology of three-week-old plants of the indicated genotypes treated with 1 μM Dex or water (-Dex) by soil drenching every other day after transplanting. **h, i** SAR induction-triggered systemic accumulation of eNAD(P) in *Dex:FIN4/fin4-3* plants with lower leaves being pretreated with or without Dex. Three lower leaves on each plant were infiltrated with 50 μM Dex or 0.1% methanol (-Dex). Twenty-four hr later, biological induction of SAR was conducted. The upper systemic leaf was collected 48 h later and AWFs were then extracted. Values are expressed relative to the eNAD(P) levels in the -Dex/-SAR samples, which were arbitrarily set to 1. Bars represent means ± SE (*n* = 3 independent AWF samples). SAR induction induced the movement of significant amounts of eNAD(P) from the Dex-treated lower leaves to the upper systemic leaves of the *Dex:FIN4/fin4-3* plants (one-way ANOVA with Tukey's test). The experiment was repeated with similar results. **j, k** Biological induction of SAR in the indicated genotypes with lower leaves being pretreated with or without Dex. Three lower leaves on each plant were infiltrated with 50 μM Dex or 0.1% methanol (-Dex). Twenty-four hr later, biological induction of SAR was determined. Bars represent means ± SE (*n* = 8 independent leaf disks). Different letters denote significant differences (one-way ANOVA with Tukey's test; *p* values are shown in the Source Data file). The photo (**k**) was taken 72 hpi. The experiment was performed three times with similar results.

background by Dex application. Based on the induction of the GUS reporter, two homozygous *Dex:FIN4/fin4-3* transgenic lines with strong induction and no leakage of GUS activity were selected for further investigation (Fig. 4f). The transgenic plants were morphologically indistinguishable from the *fin4-3* mutant and accumulated similarly reduced NAD levels as in *fin4-3* (Fig. 4g and Supplementary Fig. 9). Dex treatment restored the morphology (Fig. 4g) and NAD levels of the *Dex:FIN4/fin4-3* plants but had no effects on *fin4-3* or wild-type plants (Supplementary Fig. 9), indicating that the expression of the *FIN4* transgene in the selected lines is tightly controlled by the inducible promoter. The transgenic plants were then used for SAR induction with Dex application in lower leaves, thus restoring NAD levels only in these leaves. The rationale was that any increase in systemic eNAD(P) levels in the transgenic plants would represent transportation of eNAD(P) from the lower leaves because these plants, like *fin4-3*, accumulate drastically reduced NAD levels in the absence of Dex treatment (Supplementary Fig. 9). When Dex was infiltrated into the lower leaves followed by SAR induction 24 h later, this triggered accumulation of significantly higher eNAD(P) levels in the systemic leaves of the *Dex:FIN4/fin4-3* plants than the control treatment (Fig. 4h, i). It has previously been shown that pathogen infection leads to eNAD(P) accumulation in the inoculated leaves[22], which is likely due to pathogen-caused cell damage. These results indicate that SAR induction results in accumulation of eNAD(P) in the lower leaves, which is then transported to upper systemic leaves. Furthermore, induction of *FIN4* in the lower leaves by Dex largely, but not completely, complemented the SAR phenotype of the *Dex:FIN4/fin4-3* plants (Fig. 4j, k). Taken together, these results indicate that eNAD(P) accumulated in primarily infected leaves can move to systemic leaves to trigger SAR and suggest that endogenous eNAD(P) is a mobile SAR signal.

### Mobile signals trigger de novo eNAD(P) accumulation in systemic tissues
The result that induction of *FIN4* in lower leaves of *Dex:FIN4/fin4-3* plants only partially complemented their SAR phenotype suggested that the FIN4-mediated NAD biosynthesis in systemic leaves might also play a role in SAR. Indeed, when *FIN4* was expressed in Dex-treated upper systemic leaves of *Dex:FIN4/fin4-3* plants, SAR induction also triggered accumulation of significantly higher eNAD(P) levels in these systemic leaves than the control treatment (Fig. 5a, b). This surprising result suggests that other mobile signals produced in the lower leaves moved to the systemic leaves where they triggered de novo eNAD(P) accumulation. Furthermore, induction of *FIN4* in the upper systemic leaves by Dex treatment almost completely complemented the SAR

phenotype of the *Dex:FIN4/fin4-3* plants (Fig. 5c, d), suggesting that de novo eNAD(P) accumulation in systemic leaves plays a central role in the establishment of SAR.

### NHP is necessary and sufficient for triggering systemic eNAD(P) accumulation
NHP has recently been shown to be a potential SAR mobile signal[14,15]. We have shown that exogenous NAD(P)⁺ can induce both local and systemic immunity in *ald1* and *fmo1*[16], suggesting that eNAD(P) might function downstream of NHP. Indeed, NHP treatment induced significant accumulation of eNAD(P) in both the treated and upper systemic leaves (Fig. 6a–d), and the total NAD(P) levels did not change significantly (Supplementary Fig. 10a–d). We then tested if local application of NHP could induce de novo eNAD(P) accumulation in systemic leaves using the *Dex:FIN4/fin4-3* transgenic plants. Infiltration of NHP into lower leaves induced accumulation of significantly higher eNAD(P) levels in the upper systemic leaves than the control treatment when NAD biosynthesis was restored in the systemic leaves by Dex application (Fig. 6e, f). These results indicate that NHP and/or NHP-induced mobile signals not only induce eNAD(P) accumulation in treated leaves, but also trigger de novo systemic eNAD(P) accumulation. Furthermore, while eNAD(P) levels in the local *Psm*-infected leaves of *fmo1* were not significantly different from those in the wild type (Supplementary Fig. 11a, b), eNAD(P) levels in the systemic leaves of *fmo1* during biological induction of SAR were significantly lower than those in the wild type (Fig. 6g, h). Since the *fmo1* mutant is completely SAR defective[31], these results indicate that NHP-mediated systemic eNAD(P) accumulation plays a crucial role in the establishment of SAR.

### NHP induces systemic immunity through eNAD(P)
After establishing that NHP induces eNAD(P) accumulation, we asked whether NHP-induced systemic immunity depends on eNAD(P) and its perception at the plasma membrane. We first tested NHP-induced systemic immunity in two previously generated *35S:CD38* transgenic lines that express the human NAD(P)-hydrolyzing ectoenzyme CD38[32]. As eNAD(P) levels are only slightly reduced in *35S:CD38* transgenic plants and the *35S:CD38* transgenic plants exhibit partially compromised SAR[32], we used a lower concentration (0.25 mM) of NHP that is able to induce significant systemic immunity in wild type. NHP-induced systemic immunity was significantly inhibited in the *35S:CD38* transgenic lines (Fig. 7a). We then examined NHP-induced systemic immune responses including defense gene expression and disease resistance in the *fin4-3*, *lecrk-VI.2*, and *bak1-5 bkk1* mutants[16,24,33]. Except for *PR1* in

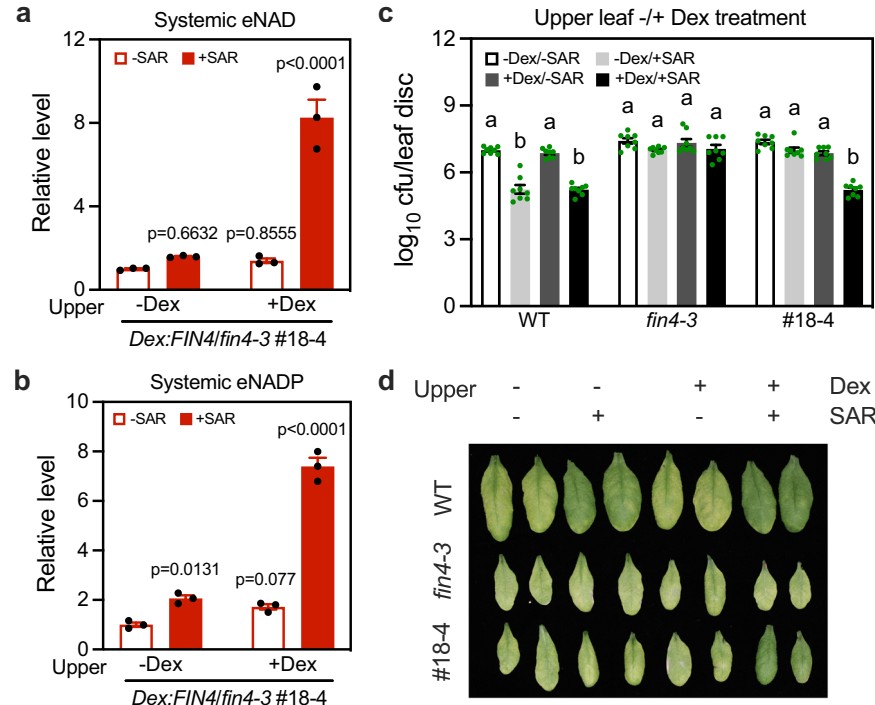

**Fig. 5 | SAR induction triggers de novo eNAD(P) accumulation in systemic tissues. a, b** SAR induction-triggered systemic accumulation of eNAD(P) in *Dex:FIN4/fin4-3* plants with upper systemic leaves being pretreated with or without Dex. One upper systemic leaf on each plant was infiltrated with 50 μM Dex or 0.1% methanol (-Dex). Twenty-four hr later, three lower leaves on each plant were infiltrated with 1 mM MgCl₂ (-SAR) or *Psm* (+SAR). AWFs from the upper systemic leaves were collected 48 h later. Values are expressed relative to the eNAD(P) levels in the -Dex/-SAR samples, which were arbitrarily set to 1. Bars represent means ± SE (*n* = 3 independent AWF samples). SAR induction induced significant eNAD(P) accumulation in the Dex-treated upper systemic leaves of the *Dex:FIN4/fin4-3* plants (one-

way ANOVA with Tukey's test). The experiment was performed three times with similar results. **c, d** Biological induction of SAR in the indicated genotypes with upper systemic leaves being pretreated with or without Dex. One upper systemic leaf on each plant was infiltrated with 50 μM Dex or 0.1% methanol (-Dex). Twenty-four hr later, biological induction of SAR was induced. Bars represent means ± SE (*n* = 8 independent leaf disks). Biological SAR was significantly induced in the Dex-treated upper systemic leaves of the *Dex:FIN4/fin4-3* plants (one-way ANOVA with Tukey's test, *p* values are shown in the Source Data file). The photo (**d**) was taken 72 hpi. The experiment was performed three times with similar results.

*lecrk-VI.2*, NHP-induced expression of *PR1*, *ALD1*, and *FMO1* as well as NHP-induced resistance to *Psm* were significantly reduced in all three mutants (Fig. 7b–d). Taken together, these results demonstrate that eNAD(P) functions downstream of NHP in activating systemic immunity.

**NHP triggers eNAD(P) accumulation through ROS**
ROS have been implicated in the induction of SAR[20,21]. Since ROS can oxidize cell membrane, leading to pore formation[34,35], they might cause NAD(P) leakage. Surely, methyl viologen (MV) treatment, which triggers ROS production by catalyzing the transfer of electrons from photosystem I to molecular oxygen[36], significantly increased eNAD(P) levels in the treated leaves (Fig. 8a). To establish a cause-and-effect relationship between ROS and eNAD(P), we measured eNAD(P) levels in systemic leaves of the *Arabidopsis* ROS deficient mutant *rbohF* during biological induction of SAR[37]. SAR induction-triggered systemic eNAD(P) accumulation was significantly reduced in the *rbohF* mutant (Fig. 8b, c). In line with this result, the *rbohF* mutant is defective in SAR[21]. We then tested whether NAD(P)⁺ treatment could restore SAR in *rbohF* and the other ROS deficient mutant *rbohD*[37]. NAD(P)⁺ treatment induced similar levels of systemic immunity in the wild type and the ROS mutants (Fig. 8d, e). These results suggest that eNAD(P) functions downstream of ROS in activating systemic immunity.

Next, we tested if NHP could induce ROS accumulation. The NHP precursor, Pip, has been shown to induce ROS accumulation in *Arabidopsis* leaves[9]. NHP also induced ROS accumulation after being infiltrated into the *Arabidopsis* leaves (Fig. 8f). To test if NHP-induced systemic eNAD(P) accumulation depends on ROS, we treated *rbohF*

with 0.5 mM NHP and measured systemic eNAD(P) levels 24 h later. Compared with that in the wild type, NHP-induced systemic accumulation of eNAD(P) was significantly inhibited in the *rbohF* mutant (Fig. 8g, h). Furthermore, NHP-induced systemic immunity was significantly weakened in both *rbohF* and *rbohD* (Fig. 8i, j). These results indicate that NHP induces systemic eNAD(P) accumulation and systemic immunity largely through RBOH-produced ROS.

## Discussion
Despite extensive studies of the SAR signaling pathway, particularly mobile signals, precisely how SAR is activated remains elusive. Here, we show that eNAD(P) accumulates systemically and triggers SAR through its receptor LecRK-VI.2 in systemic tissues. Although a direct movement of endogenous eNAD(P) is difficult to demonstrate, our data obtained using the *Dex:FIN4/fin4-3* transgenic plants in which NAD biosynthesis can be spatially controlled by Dex application provide compelling support for the idea that systemic eNAD(P) originates from two sources: movement of eNAD(P) through the apoplastic route to systemic tissues as a potential mobile signal and de novo eNAD(P) accumulation in systemic tissues triggered by other mobile signals. These results suggest that SAR mobile signals might converge on eNAD(P) in systemic tissues to trigger SAR.

Indeed, infiltration of NHP into lower leaves induced eNAD(P) accumulation in upper systemic leaves (Fig. 6e, f). Interestingly, although the NHP biosynthesis mutant *fmo1* accumulates significantly reduced eNAD(P) levels in systemic leaves upon SAR induction (Fig. 6g, h), eNAD(P) levels in the inoculated local leaves are comparable in *fmo1* and the wild type (Supplementary Fig. 11a, b). Since SAR is completely

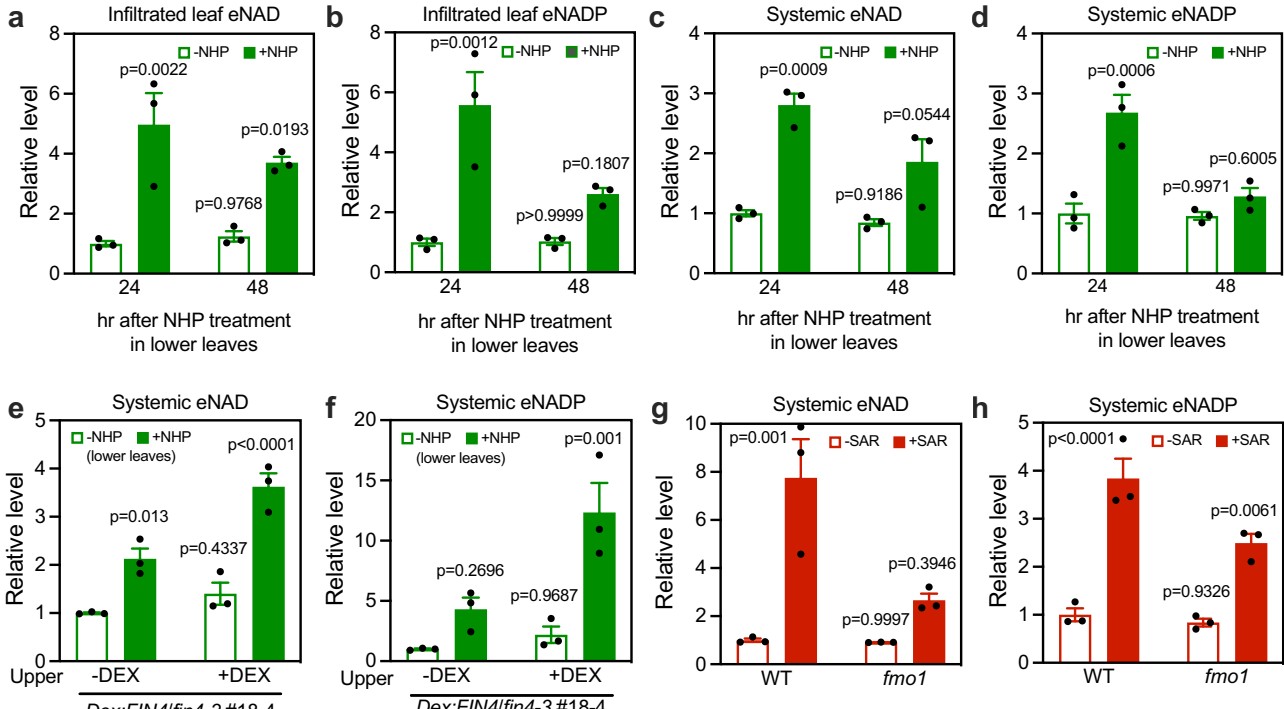

**Fig. 6 | NHP induces de novo eNAD(P) accumulation in systemic tissues.**
**a**–**d** eNAD(P) levels in NHP-infiltrated leaves (**a** and **b**) or upper systemic leaves (**c** and **d**). For (**a** and **b**), two leaves on each wild-type plant were infiltrated with 0.5 mM NHP or water (-NHP). For (**c** and **d**), three lower leaves on each wild-type plant were infiltrated with 0.5 mM NHP or water. The infiltrated leaves (**a** and **b**) or one systemic leaf (**c** and **d**) on each plant was collected at the indicated times and then AWFs were extracted. Values are expressed relative to the eNAD(P) levels in the -NHP samples at 24 h, which were arbitrarily set to 1. Bars represent means ± SE ($n = 3$ independent AWF samples). NHP induced significant eNAD(P) accumulation in both the infiltrated and the upper systemic leaves (one-way ANOVA with Tukey's test). The experiments were repeated with similar results. **e**, **f** NHP-induced systemic accumulation of eNAD(P) in *Dex:FIN4/fin4-3* plants with systemic leaves being pretreated with or without Dex. One upper systemic leaf on each plant was infiltrated with 50 µM Dex or 0.1% methanol (-Dex). Twenty-four hr later, three lower leaves on the plant were infiltrated with 0.5 mM NHP or water (-NHP). Twenty-four hr later, the pretreated systemic leaves were collected. Values are expressed relative to the eNAD(P) levels in the -Dex/-NHP samples, which were arbitrarily set to 1. Bars represent means ± SE ($n = 3$ independent AWF samples). The NHP treatment of the lower leaves induced significant eNAD(P) accumulation in the Dex-treated upper systemic leaves of the *Dex:FIN4/fin4-3* plants (one-way ANOVA with Tukey's test). The experiment was repeated with similar results. **g**, **h** SAR induction-triggered systemic accumulation of eNAD(P) in wild type (WT) and *fmo1*. Values are expressed relative to the eNAD(P) levels in the WT/-SAR samples, which were arbitrarily set to 1. Bars represent means ± SE ($n = 3$ independent AWF samples). The SAR induction-induced systemic eNAD(P) accumulation was significantly reduced in *fmo1* (one-way ANOVA with Tukey's test). The experiments were repeated with similar results.

absent in the *fmo1* mutant[31], these results suggest that eNAD(P) originating from the local leaves is not sufficient for activating effective SAR and that de novo eNAD(P) accumulation in systemic tissues triggered by NHP and possibly other FMO1-dependent mobile signals is essential for SAR establishment. Consistently, NHP-induced systemic immunity largely depends on eNAD(P) and its receptor complex LecRK-VI.2/BAK1 (Fig. 7). These results, together with our previous finding that NAD(P)+ can induce wild-type levels of systemic immunity in the NHP biosynthesis mutants, *ald1* and *fmo1*[16], strongly suggest that NHP functions upstream of the eNAD(P)-LecRK-VI.2/BAK1 signaling module by triggering eNAD(P) accumulation in systemic tissues.

We found that NHP triggers eNAD(P) accumulation mainly through ROS production. The NHP precursor Pip has been shown to induce ROS accumulation[9], whereas the NHP biosynthesis mutants, *ald1* and *fmo1*, fail to accumulate ROS in response to pathogen infection and excess light stress, respectively[9,38]. Moreover, ROS can induce ion leakage in paraquat-damaged maize leaves[39], while *ald1* and the ROS deficient *rboh* mutants exhibited reduced ion leakage upon pathogen infection[9,37]. In accordance with these results, NHP treatment results in ROS accumulation (Fig. 8f) and ROS accumulation triggers eNAD(P) accumulation (Fig. 8a). More importantly, NHP-induced eNAD(P) accumulation and systemic immunity are significantly inhibited in the ROS deficient *rbohF* mutant (Fig. 8g–j), indicating that NHP induced eNAD(P) accumulation primarily through RBOHF-generated

ROS. However, how NHP activates RBOHF during the induction of SAR requires further investigation.

ROS have been proposed to mediate a reiterative signal network that underlies the establishment of SAR[20]. It has been shown that inoculation of lower leaves on *Arabidopsis* plants with the avirulent pathogen *Pst avrRpt2* induces strong oxidative bursts and micro-oxidative bursts in the inoculated and upper systemic leaves, respectively, and that preventing ROS production either in the inoculated or systemic leaves with the NADPH oxidase inhibitor diphenyleneiodonium compromised the establishment of SAR[20]. These results indicate that ROS produced by the NADPH oxidases (e.g., RBOHD and RBOHF) in both primarily infected and systemic leaves are required for SAR. A recent report confirmed accumulation of ROS in systemic leaves during SAR induction and provided convincing genetic evidence for the pivotal role of ROS in SAR[21]. It was proposed that ROS might act additively to generate AzA or its precursor 9-oxononanoic acid[21]. Our results corroborate the importance of ROS in SAR and uncovered a novel mechanism whereby ROS induce SAR signaling. We found that ROS trigger eNAD(P) accumulation (Fig. 8a) and that accumulation of eNAD(P) in systemic leaves of the ROS deficient mutant *rbohF* is significantly inhibited during the induction of SAR (Fig. 8g, h). These results indicate that, like NHP, ROS are also both necessary and sufficient for triggering eNAD(P) accumulation in systemic tissues.

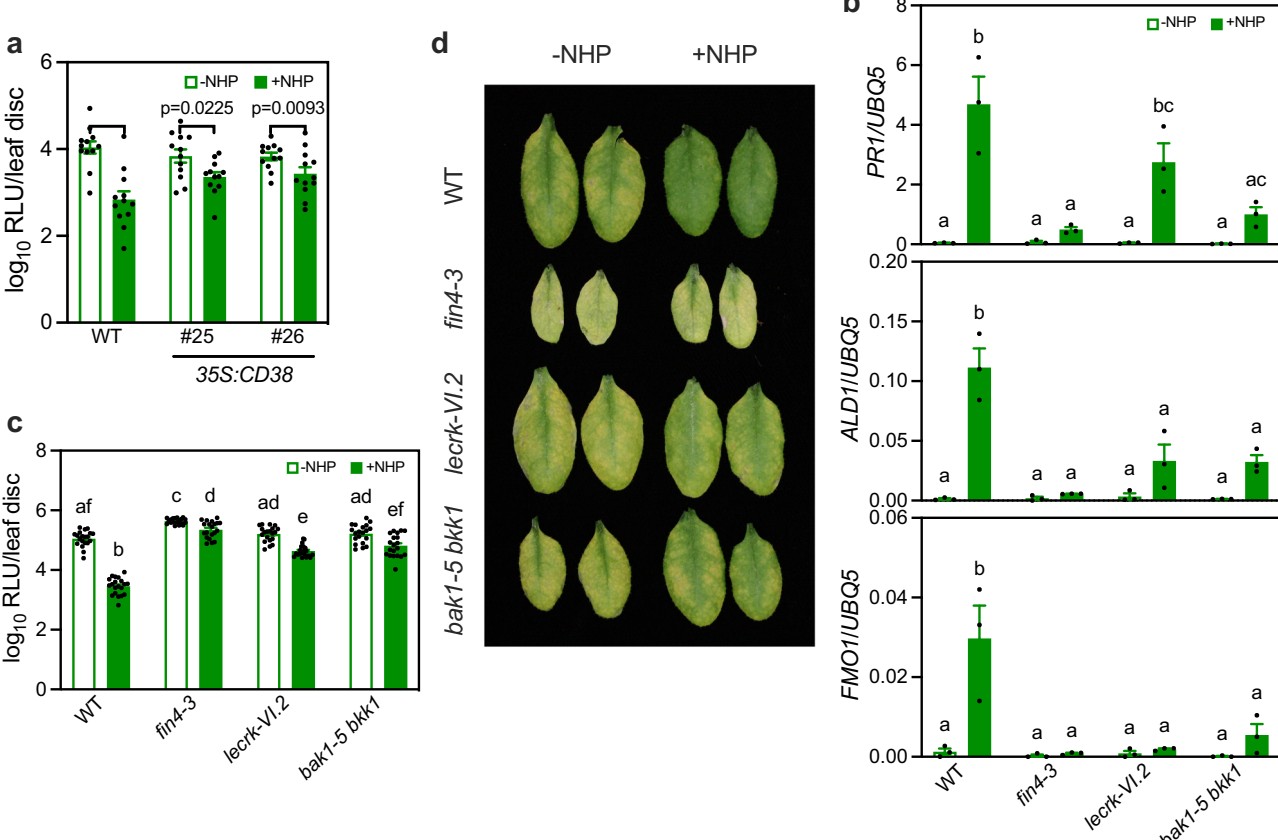

**Fig. 7 | NHP-induced systemic immunity depends on eNAD(P) and its receptor complex. a** NHP-induced systemic immunity in WT and *35 S:CD38* plants. Three lower leaves were infiltrated with 0.25 mM NHP or water (-NHP). Twenty-four hr later, one upper systemic leaf on each plant was inoculated with *Psm*. Samples were collected at 72 hpi. Bars represent means ± SE (*n* = 12 independent leaf disks). The NHP-induced systemic immunity was significantly reduced in the *35 S:CD38* trans-genic plants (two-way ANOVA with Sidak's test). The experiment was performed three times with similar results. **b** NHP-induced systemic expression of *PR1*, *ALD1*, and *FMO1* in the indicated genotypes. Three lower leaves on each plant were infiltrated with 0.5 mM NHP or water (-NHP). One upper systemic leaf on each plant was collected 24 h later. Bars represent means ± SE (*n* = 3 independent total RNA samples). Different letters denote significant differences (one-way ANOVA with Tukey's test; *p* values are shown in the Source Data file). The experiment was repeated with similar results. **c**, **d** NHP-induced systemic immunity in the indicated genotypes. The photo in (**d**) was taken at 72 hpi. Bars in (**c**) represent means ± SE (*n* = 20 independent leaf disks). Different letters denote significant differences (one-way ANOVA with Tukey's test; *p* values are shown in the Source Data file). The experiment was performed three times with similar results.

Although the relationship between eNAD(P) and other potential SAR signals remains to be consolidated, available evidence supports the idea that eNAD(P) serves as a converging point of SAR signals in systemic tissues. We have shown that NAD(P)[+] treatment induces wild-type levels of immunity in *dir1-1*[6,16], the AzA signaling mutant *azi1*[11], and the G3P biosynthesis mutants *sfd1* (also called *gly1*) and *nho1* (also called *gli1*)[16,40–42], suggesting that eNAD(P) may function downstream of DIR1, AzA, and G3P. Whereas it is unclear how G3P acts in systemic leaves, the G3P mutants, *gly1 gli1* and *gly1*, accumulate significantly reduced levels of Pip and monoterpenes, respectively[9,43], suggesting that G3P boosts Pip and monoterpene production in systemic leaves. Moreover, Pip, monoterpenes, and NO promote ROS accumulation[9,17,21], and conversely, ROS is required for Pip accumulation in systemic leaves[9]. SA also elevates ROS accumulation[44–46], and NAD(P)-induced immunity depends on SA[22]. Thus, it is thought that these SAR signals form a signaling amplification loop in systemic tissues[9,47–49]. Since ROS is short-lived[50], and eNAD(P) induces short-duration systemic immunity (Supplementary Fig. 1a–c), the function of the signaling amplification loop in systemic leaves is likely to prompt prolonged production of ROS for triggering sustained eNAD(P) accumulation, which in turn induces persistent SAR signaling.

Combining our results and others, we propose a working model to illustrate how SAR is induced (Fig. 8k). At primary infection sites, PTI, DTI, and ETI responses cause production of mobile signals including eNAD(P) and NHP[22,51,52], which are subsequently trans-ported to systemic tissues where eNAD(P) binds to and activates its receptor complex LecRK-VI.2/BAK1. The contribution of the initial eNAD(P)-LecRK-VI.2/BAK1 signaling to SAR appears to be limited, since systemic movement of eNAD(P) in the *fmo1* mutant is not suf-ficient for activating effective SAR. On the other hand, NHP and other mobile signals trigger a signaling amplification loop, leading to accumulation of ROS that induce de novo eNAD(P) accumulation. The new eNAD(P) further activates the receptor complex to boost the strength of SAR signaling, triggering the downstream SA/NPR1-dependent immune responses[15,16,53,54]. eNAD(P) can also promote the biosynthesis of SA and NHP in systemic tissues by upregulating their biosynthesis genes[55]. Since none of the mutants used in this study, including *rbohD*, *rbohF*, *fin4-3*, *lecrk-VI.2*, and *bak1-5 bkk1*, can com-pletely block NHP-induced systemic immunity (Figs. 7c, 8i), a modest NHP-dependent but ROS- and eNAD(P)-LecRK-VI.2/BAK1-indepen-dent pathway to SAR seems to exist. Nevertheless, based on our findings, we suggest that SAR is essentially a systemic immune response triggered by various mobile signals that converge largely on the eNAD(P)-receptor module. In contrast, PTI, DTI, and ETI are triggered by large numbers of diverse PAMPs, DAMPs, and effectors, each with a specific receptor. It seems that at the primary infection site, plants must evolve large numbers of PRRs and NLRs to combat rapidly changing plant pathogens, whereas in distal systemic tissues

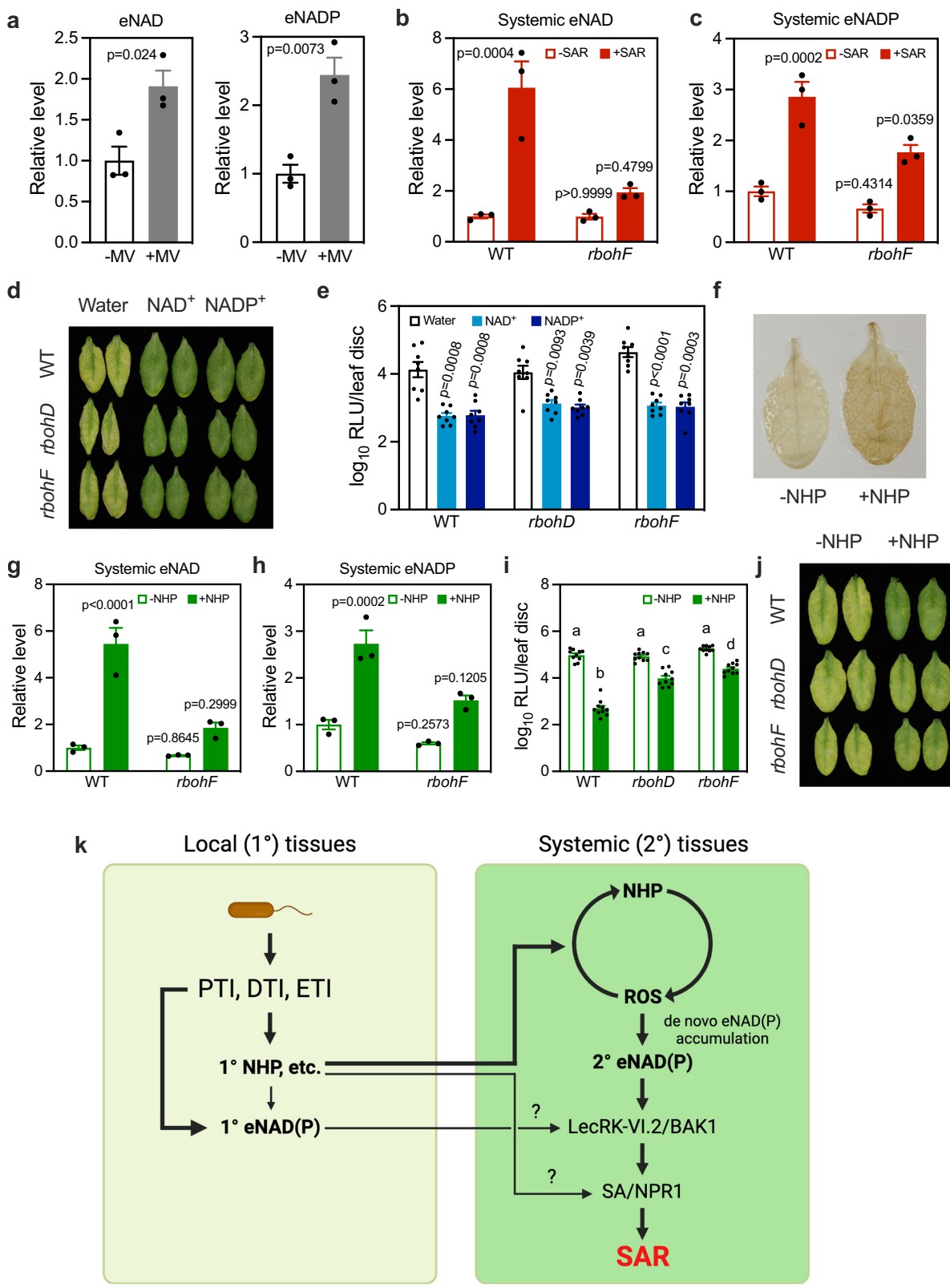

where selective pressures are absent, the presence of NAD(P), a conserved coenzyme, in the extracellular space is sufficient to prime immunity. Our findings thus clarify the induction mechanism of SAR, allowing for a holistic understanding of the plant immune system. Future work will be focused on unraveling the eNAD(P) signaling network in model and crop plants to help develop durable crop protection strategies.

## Methods

### Bacterial strains

*E. coli* XL1-blue was cultured at 37 °C in LB medium supplemented with appropriate antibiotics for plasmid extraction. *Agrobacterium tumefaciens* GV3101(pMP90) was cultured at 28 °C in LB medium supplemented with appropriate antibiotics for *Arabidopsis* transformation. *Psm*, *Psm_lux*, and *Pst avrRpt2* were cultured at 28 °C in King's B

**Fig. 8 | NHP treatment triggers eNAD(P) accumulation through RBOHF-generated ROS. a** eNAD(P) levels in the leaves treated with or without MV. Values are expressed relative to the eNAD(P) levels in the -MV samples, which were arbitrarily set to 1. Bars represent means ± SE (*n* = 3 independent AWF samples). The MV treatment induced significant eNAD(P) accumulation (two-tailed *t* test). The experiment was repeated with similar results. **b, c** SAR induction-triggered systemic accumulation of eNAD(P) in wild type (WT) and *rbohF*. Values are expressed relative to the eNAD(P) levels in the WT/-SAR samples, which were arbitrarily set to 1. Bars represent means ± SE (*n* = 3 independent AWF samples). The SAR induction-induced eNAD(P) accumulation was significantly inhibited in *rbohF* (one-way ANOVA with Tukey's test). The experiment was performed three times with similar results. **d, e** NAD(P)⁺-induced systemic immunity in the indicated genotypes. The photo in (**d**) and samples in (**e**) were taken at 72 hpi. Bars in (**e**) represent means ± SE (*n* = 8 independent leaf disks). NAD(P)⁺ induced similar levels of systemic immunity in the WT, *rbohD*, and *rbohF* plants (two-tailed *t* test). The experiment was performed three times with similar results. **f** DAB (3, 3′-diaminobenzidine) staining of NHP-induced H₂O₂. *Arabidopsis* leaves were infiltrated with 0.5 mM NHP or water (-NHP) and collected 24 h later. **g, h** NHP-induced systemic accumulation of eNAD(P) in WT and *rbohF*. Values are expressed relative to the eNAD(P) levels in WT/-NHP samples, which were arbitrarily set to 1. Bars represent means ± SE (*n* = 3 independent AWF samples). The NHP-induced eNAD(P) accumulation was

significantly inhibited in *rbohF* (one-way ANOVA with Tukey's test). The experiment was conducted three times with similar results. **i, j** NHP-induced systemic immunity in the indicated genotypes. The photo in (**j**) and samples in (**i**) were taken at 72 hpi. Bars in (**i**) represent means ± SE (*n* = 10 independent leaf disks). The NHP-induced systemic immunity was significantly reduced in *rbohD* and *rbohF* (one-way ANOVA with Tukey's test; *p* values are shown in the Source Data file). The experiment was performed three times with similar results. **k** A proposed working model for the function of eNAD(P) in SAR. Pathogen infections activate PTI, DTI, and ETI responses in the local leaf tissues, which are accompanied by generation of a blend of mobile signals among which are eNAD(P) and NHP. NHP likely contributes to eNAD(P) accumulation in the local leaf tissues. These mobile SAR signals are swiftly transported to systemic leaf tissues where the eNAD(P) from the local tissues binds to and activates its receptor complex LecRK-VI.2/BAK1, but this signal appears to be too weak to trigger downstream SAR signaling. Other mobile signals including NHP initiate a signaling amplification loop, triggering accumulation of ROS that induce de novo eNAD(P) accumulation in the systemic leaf tissues. The fresh eNAD(P) further activates the receptor complex to boost the force of the SAR signaling, triggering the downstream SA/NPR1-mediated SAR responses. A modest NHP-dependent but ROS- and eNAD(P)-independent pathway to SAR seems to exist. A question marks (?) indicates that the contribution of the pathway to SAR needs further investigation. The figure was created with BioRender.com.

medium (2% proteose peptone, 0.15% K₂HPO₄, 6 mM MgSO₄, and 1.5% glycerol) containing appropriate antibiotics. *Psm*, *Psm_lux*, and *Pst avrRpt2* in overnight log-phase cultures were centrifuged at 3000 g for 5 min to collect cells. Pellet was resuspended in 1 mM MgCl₂ and diluted to different OD₆₀₀ concentrations for leaf inoculation experiments.

### Arabidopsis

*Arabidopsis* plants were grown in soil in a growth room for 3–4 weeks before experiments at 24/22 °C (day/night) and 60% relative humidity with a 14/10 h light/dark photoperiod. To grow *Arabidopsis* seedlings on MS medium, seeds were first surface sterilized with 15% (v/v) bleach and washed thoroughly in sterile water twice, and then germinated on sterile half-strength (½) MS solid medium (pH 5.7) supplemented with 1% sucrose and 0.6% agar with appropriate antibiotics. Plated seedlings were grown in a growth chamber at 24/22 °C (day/night) with a 16/8 h light/dark photoperiod.

*Arabidopsis* plants used in this study are all in wild-type (WT) Col-0 background. Previously published lines are: *lecrk-VI.2-2*[16], *bak1-5 bkk1*[33], *fin4-3*[24], *35 S:CD38*[32], *fmo1*[31], *rbohD*[37], and *rbohF*[37]. Accession numbers of these mutants are: SAIL_1146_B02 (*lecrk-VI.2-2*), SAIL_1145_B10 (*fin4-3*), and SALK_026163 (*fmo1*). *Arabidopsis* transgenic lines generated in this study are described below. All transgenic lines used in this study are single insertion homozygous plants.

### Plasmid construction and plant transformation

To complement the *fin4-3* mutant, the *35 S:FIN4* construct was generated. The full length CDS (1956 bp) of *FIN4* was amplified from wild-type cDNAs using the oligos KpnI-FIN4F and XbaI-FIN4R (Supplementary Table 1). The PCR products were digested with *Kpn*I and *Xba*I and cloned into the *kpn*I/*Xba*I-digested pCAMBIA1300S vector to create pCAMBIA1300S-FIN4. To make *Dex:FIN4/fin4-3* and *Dex:LecRK-VI.2/lecrk-VI.2* plants, full length CDSs of *FIN4* (1956 bp) and *LecRK-VI.2* (2049 bp) were amplified from wild-type cDNAs using oligos attB1-FIN4F/attB1FIN4R and attB1LecRK-VI.2 F/attB1LecRK-VI.2 R (Supplementary Table 1), respectively. PCR products were used as templates for the second round of PCR with the oligos Adapter attB1 and Adapter attB2 to add attB1 and attB2 sites to the ends. The resulting PCR products were cloned into the pDONR221 vector by Gateway BP reactions to obtain the entry vectors pDONR221-FIN4 and pDONR221-LecRK-VI.2. Gateway LR reactions between the entry vectors and the pOpON vector were then performed to create pOpON-FIN4 and pOpON-LecRK-VI.2. All constructs were confirmed by sequencing.

The *A. tumefaciens* strain GV3101(pMP90) was transformed with the binary constructs carrying the indicated transgenes. The floral dip method was used to transform *Arabidopsis* plants[56]. T₁ transgenic plants were selected on ½ MS plates containing appropriate antibiotics. Single insertion and homozygous lines were selected in the T₂ and T₃ generations, respectively.

### Generation of bioluminescent Psm_lux

*Psm_lux* was generated as previously described[57]. Briefly, *Psm* was transformed with the pBJ2 plasmid and selected on solid King's B medium containing streptomycin and gentamycin at 28 °C for 48–72 h. Transformants were cultured in liquid King's B medium containing 0.1% L-arabinose overnight at 28 °C to induce transposition. The resulting culture was streaked onto King's B plates with streptomycin and incubated at 28 °C for 48–72 h until single colonies appeared. Bioluminescent colonies were patched onto King's B plates with gentamycin to select gentamicin-sensitive colonies, which confirmed the absence of the pBJ2 plasmid. The insertion of *luxCDABE* into the attTn7 site was confirmed by PCR and sequencing with the oligos glmS_pstF and PTn7R[57].

### NA, NAD⁺, and NADP⁺ treatment

NA, NAD⁺, and NADP⁺ were dissolved in ddH₂O to make solutions with appropriate concentrations. The pH of NA, NAD⁺, and NADP⁺ solutions was adjusted to 5.7 with NaOH. For all the experiments in this paper, NA, NAD⁺, and NADP⁺ solutions were freshly made before the experiment. For NA-, NAD⁺-, and NADP⁺-induced local resistance, 4 mM NA (unless otherwise indicated), 0.2 mM NAD⁺ (unless otherwise indicated), or 0.4 mM NADP⁺ (unless otherwise indicated) was infiltrated into two fully expanded leaves per plant (the 5th and 6th from the bottom) using 1-mL needleless syringe from the abaxial side of leaves. Water was infiltrated in the same manner as the control. After 4 h (unless otherwise indicated), the infiltrated leaves were either inoculated with *Psm* or *Psm_lux* (OD₆₀₀ = 0.001) by infiltration or collected for RNA extraction and total NAD measurement. Resistance to *Psm* was assessed 72 h post inoculation (hpi).

For NA-, NAD⁺-, and NADP⁺-induced systemic immunity, 4 mM NA (unless otherwise indicated), 1 mM NAD⁺ (unless otherwise indicated), or 1 mM NADP⁺ (unless otherwise indicated) was infiltrated into three lower leaves per plant (the 3rd, 4th, and 5th from the bottom). Water was infiltrated in the same way as the control. After 4 h (unless otherwise indicated), one upper systemic leaf (the 6th from the bottom) was either inoculated with *Psm* or *Psm-lux* (OD₆₀₀ = 0.001) by

infiltration or collected for RNA extraction. Resistance to *Psm* was assessed 72 hpi.

## Biological induction of SAR

Three lower leaves (the 3rd, 4th, and 5th from bottom) on each four-week-old *Arabidopsis* plant were infiltrated with 1 mM MgCl₂ (-SAR), *Psm* ($OD_{600}$ = 0.002) (SAR), or *Pst Rpt2* ($OD_{600}$ = 0.002). After 48 h, one upper systemic leaf (the 6th from bottom) on each plant was either collected for RNA extraction or inoculated with *Psm* or *Psm-lux* ($OD_{600}$ = 0.001) by infiltration. *Psm* titers were assessed 72 hpi. In some experiments, the systemic leaves were collected at the specified times for eNAD and eNADP measurement after SAR induction.

## Basal resistance to Psm

To determine basal resistance to *Psm*, two leaves (the 5th and 6th from the bottom) on each four-week-old plant were infiltrated with *Psm* ($OD_{600}$ = 0.0001). Titers were determined 72 hpi.

## Quantification of Psm growth

*Psm* growth was quantified using either the traditional colony-forming unit (CFU) method or a bioluminescence method[57]. For the CFU method, leaf disks (7 mm in diameter) were taken at 72 hpi using a hole punch, placed into 500 μL of 1 mM MgCl₂ in a 1.5-mL microcentrifuge tube, and ground forcefully using a plastic pestle. Twenty-fold serial dilutions of the homogenate were plated on Trypticase Soy Agar (TSA) medium supplemented with 50 μg/mL streptomycin to determine the bacterial titers. CFU was calculated by multiplying the numbers of colonies by the dilution factors. Bacterial titers were expressed as $log_{10}$(CFU) per leaf disk. One leaf disc was taken from each leaf.

For the bioluminescence method, leaf disks (7 mm in diameter) were collected using a hole punch and placed in a white, light-reflecting 96-well plate (Corning, Cat#3912). One leaf disc was taken from each leaf. Leaf disks were floated on 150 μL 1 mM MgCl₂ in each well to keep them wet. The plate containing leaf disks was placed in the sample drawer of a GloMax Discover luminometer (Promega) and kept in the dark by closing the lid for 10 min to reduce background signals. The relative light unit (RLU) of each sample was then measured for 10 s. Bacterial titers were expressed as $log_{10}$(RLU) per leaf disk.

## Dexamethasone treatment

Dexamethasone (Dex) was dissolved in methanol to make a 50 mM stock solution and preserved at -20 °C. To induce the expression of target genes, 50 μM Dex (1000-fold ddH₂O dilution of the stock solution) was infiltrated into lower leaves (the 3rd, 4th, and 5th from the bottom) or upper leaves (the 6th from the bottom) using a 1 mL-needleless syringe. Methanol (0.1%) was used as the solvent control. The target gene is generally induced within several hr and the induction can last for several days[30]. All downstream experiments were performed 24 h after Dex treatment. To rescue the mutant morphology, 1 μM Dex was applied through soil drenching every other day for 4 weeks. Methanol (0.02%) was used as the mock control. Photos of the plants were taken at the end of the experiment.

## NHP treatment

NHP powder was dissolved in ddH₂O to make a 0.5 mM solution. The NHP solution was aliquoted and frozen at −20 °C. For NHP induced local responses, 0.5 mM NHP or ddH₂O (-NHP) was infiltrated using a 1 mL-needleless syringe into two leaves (the 5th and 6th from the bottom) on each four-week-old plant. After 24 h (unless otherwise indicated), the treated leaves were collected for indicated assays. For NHP induced systemic responses, 0.5 mM NHP or ddH₂O was infiltrated into three lower leaves (the 3rd, 4th, and 5th from the bottom) on each four-week-old plant. After 24 h (unless otherwise indicated), one upper systemic leaf (the 6th from the bottom) was collected for indicated assays. For NHP induced systemic immunity, 0.5 mM NHP

(unless otherwise indicated) or ddH₂O was infiltrated into three lower leaves (the 3rd, 4th, and 5th from the bottom) on each four-week-old plant. After 24 h, one upper leaf was inoculated with *Psm_lux* ($OD_{600}$ = 0.001) by infiltration. *Psm* titers were assessed 72 hpi.

## GUS staining and DAB staining

Dex-treated leaves were stained for GUS activity as previously described[58]. Briefly, leaves were submerged in a solution containing 1 mg/mL X-Gluc in 50 mM Na₂HPO₄ pH 7.0, 10 mM EDTA, 0.5 mM potassium ferricyanide, 0.5 mM potassium ferrocyanide, and 0.06% Triton X-100, and vacuum infiltrated for 5 min. The staining solution was removed after overnight incubation at 37 °C, and the samples were cleared of chlorophyll by sequential changes of 75% and 95% ethanol. For DAB (3, 3′-diaminobenzidine) staining, leaves were immersed in a DAB solution (1 mg/mL, pH 3.8) overnight, and then boiled in ethanol for 10 min followed by several washes in ethanol.

## RNA extraction and quantitative PCR (qPCR)

Total RNA was extracted as previously described[27]. Briefly, 100 mg leaf tissues were snap-frozen in liquid nitrogen, ground to a fine powder in a 2 mL lysing tube with a Spex SamplePrep 2000 Geno/Grinder (OPS Diagnostics), and mixed with 500 μL water-saturated phenol and 500 μL RAPD RNA extraction buffer (100 mM LiCl, 100 mM Tris-HCl pH8.0, 10 mM EDTA, and 1% SDS). After incubation at 65 °C for 5 min, the samples were centrifuged at 15,000 g for 5 min at room temperature. The aqueous phase (500 μL) was mixed with 500 μL chloroform: isoamyl alcohol (v: v, 24: 1). After centrifugation at 15,000 g for 5 min at 4 °C, the aqueous phase was transferred to a new 1.5 mL tube. RNA was precipitated by addition of an equal volume of isopropanol, incubation at room temperature for 10 min, and centrifugation at 15,000 g for 10 min at 4 °C. The RNA pellet was washed with 75% ethanol, air-dried on ice for 20 min, and dissolved in 100 μL DEPC-water. RNA concentration and quality were determined using a NanoDrop 2000 spectrometer (Thermo Scientific).

For reverse transcription (RT), total RNA was first treated with a Turbo DNA Free Kit (Invitrogen, Cat#AM1907) at 37 °C for 30 min. After inactivation of the DNase, RT was performed using SuperScript™ IV First-Strand Synthesis kit (Invitrogen, Cat#18091050) according to the user's manual. cDNA was diluted and used for qPCR. qPCR was performed using SYBR™ Green PCR Master Mix (Applied Biosystems, Cat#4309155) on a QuantStudio 3 Real-Time PCR system (Applied Biosystems) according to the user's manual. The $2^{-\Delta Ct}$ method was used to determine the relative level of gene expression. *UBQ5* was used as an internal control. The primers used for qPCR were reported previously[16,59].

## RNA-seq library construction, sequencing, and data analysis

Total RNA was cleaned with the RNeasy MiniElute Cleanup Kit (Qiagen, Cat#74204) followed by on-column DNase digestion (Qiagen, Cat#79254). RNA samples were measured by the QUBIT fluorescent method (Invitrogen) and Agilent Bioanalyzer. An amount of 250 ng of high-quality total RNA with RIN of 7 or higher was used for library construction using the reagents provided in the NEBNext Poly(A) mRNA Magnetic Isolation Module (New England Biolabs, Cat#E7490) and the NEBNext Ultra II Directional RNA Library Prep Kit (New England Biolabs, Cat#E7760) according to the manufacturer's user guide. Briefly, 200 ng of total RNA was used for mRNA isolated using the NEBNext Poly(A) mRNA Magnetic Isolation Module (New England Biolabs, Cat#E7490). The poly(A) enriched RNA was then fragmented in NEBNext First Strand Synthesis Buffer via incubation at 94 °C for 15 min. This step was followed by first-strand cDNA synthesis using reverse transcriptase and random hexamer primer. Synthesis of double-strand cDNA was done using the 2nd strand master mix provided in the kit, followed by end-repair and dA-tailing. At this point, Illumina adapters were ligated to the sample. Finally, library was

amplified, followed by purification with AMPure beads (Beckman Coulter, Cat#A63881). The library size and mass were assessed by analysis in the Agilent TapeStation using a High Sensitivity DNA1000 Screen Tape. A 250-900 library peak was observed with the highest peak at ~420 bp. Barcoded libraries were pooled equimolarly for sequencing simultaneously for NavaSeq 6000 S4 2 × 150 cycles run as described below. RNA-seq library construction was performed at The University of Florida (UF) ICBR Gene Expression Core (https://biotech.ufl.edu/gene-expression-genotyping/, RRID:SCR_019145).

Normalized libraries were submitted to the "Free Adapter Blocking Reagent" protocol (FAB, Cat#20024145) to minimize the presence of adapter-dimers and index hopping rates. The library pool was diluted to 0.8 nM and sequenced on one S4 flow cell lane (2 × 150 cycles) of the Illumina NovaSeq6000. The instrument's computer utilized the NovaSeq Control Software v1.6. Cluster and SBS consumables were v1.5. The final loading concentration of the library was 120 pM with 1% PhiX spike-in control. One lane generated 2.5–3 billion paired-end reads (~950 Gb) with an average Q30% > = 92.5% and Cluster PF = 85.4%. FastQ files were generated using the BCL2fastQ function in the Illumina BaseSpace portal. The Illumina NovaSeq 6000 was used to sequence the libraries for 2 × 150 cycles. Sequencing was performed at the UF ICBR NextGen Sequencing (https://biotech.ufl.edu/next-gen-dna/, RRID:SCR_019152).

Reads acquired from the Illumina NovaSeq 6000 platform were cleaned with the Cutadapt package (v3.4)[60] for trimming adapters and low-quality bases with a quality phred-like score <20. Reads <75 pbs were excluded from RNA-seq analysis. The genome of *A. thaliana* (version TAIR10) from the database of ENSEMBL was used as the reference sequences for RNA-seq analysis. The cleaned reads of each sample were mapped to the reference genome by using the STAR package (Spliced Transcripts Alignment to a Reference, v2.7.9a)[61]. Mapping results were processed with the HTSeq (High-Throughput Sequence Analysis in Python, v0.11.2)[62], samtools, and scripts developed in house at UF ICBR to remove potential PCR duplicates and count uniquely mapped reads for gene expression analysis. PCA analysis (for detecting outlier samples) based on all identified genes in each analysis was performed with the R-package (v4.1.3). The RNA-seq was conducted with three biological replicates per genotype/treatment. The gene expression levels were analyzed by a DESeq2-based R pipeline[63]. Heatmap was made with TBtools[64]. GO analysis was conducted with PANTHER version 14 and the graphics was created with R-package ggplot2[65,66].

## Total NAD and NADP measurement

To measure total NAD and NADP levels, indicated leaves were collected, weighed, and immediately boiled in 80% ethanol at 95 °C for 3 min. The supernatant was diluted 10-fold with ddH$_2$O and used for NAD and NADP measurement with the NAD/NADH-Glo™ Assay and NADP/NADPH -Glo™ Assay kits (Promega, Cat#G9071 and G9081) according to the user's manual. All reactions were prepared on ice. Values were normalized to the leaf weight.

## eNAD and eNADP measurement

To measure eNAD and eNADP levels, apoplastic wash fluids (AWFs) were extracted based on the previously described method[67]. Briefly, indicated leaves were cut with a sharp blade, weighed, rinsed in ddH$_2$O, and vacuum-infiltrated with ddH$_2$O in a 60-mL syringe. Water on the surface of infiltrated leaves was carefully removed with Kimwipes (Kimberly-Clark Professional, Cat#34120). To collect AWF, leaves were centrifuged at 500 g for 5 min at 4 °C. The volume of the AWF was brought to 500 μL with ddH$_2$O and the solution was filtered through 3-kD MWCO filters (Sartorius, Cat#VS0192) by centrifugation at 13,000 g for 30 min at 4 °C. Filtered AWF was immediately used for NAD and NADP measurement with the NAD/NADH-Glo™ Assay and NADP/NADPH -Glo™ Assay kits according to the user's manual. AWF

was kept on ice and all reactions were prepared on ice. Values were normalized to the leaf weight.

## $^{13}$C-NAD$^+$ treatment and LC-MS/MS assay

One mg $^{13}$C-NAD$^+$ was dissolved in 14.59 mL ddH$_2$O to make a 100 μM solution. Three lower leaves (the 3rd, 4th, and 5th from the bottom) on each of 20 four-week-old wild-type plants were infiltrated with the 100 μM $^{13}$C-NAD$^+$ solution using a needleless syringe. Three biological samples each containing 20 of the infiltrated leaves were collected and weighed. The upper noninfiltrated systemic leaves were divided into three biological samples and weighed separately. Different volumes (20× the sample weight) of 80% ethanol were added into the samples. After boiling at 95 °C for 3 min, the samples were cooled down on ice. One mL of the solution from each sample was lyophilized using a lyophilizer (Labconco Corporation). Each sample was resuspended in 1 mL 80% methanol and diluted (10×) in 0.1% formic acid water solution. Ten μl of this diluted sample was injected for liquid chromatography-mass spectrometry (LC-MS/MS) analysis. LC-MS/MS was conducted using an TSQ Altis mass spectrometer and Vanquish-Flex LC (Thermo Fisher Scientific). An Accucore C18 HPLC Column (Thermo Fisher Scientific, Cat#17126-102130) with particle size of 2.6 μm, diameter of 2.1 mm, and length of 100 mm was used. Solvent A contained 0.1% (v/v) formic acid in deionized water, and solvent B had 0.1% (v/v) formic acid in acetonitrile with a flow rate of 0.2 ml/min. A 5 min gradient was used for separation in positive mode. The LC gradient starts at 0% B, then linearly increased to 50% (v/v) solvent B for 3 min, then ramped to 95% (v/v) B for 3.5 min and maintained at 95% (v/v) B over 4.2 min, followed by holding at 0% B for 4.4 and 4.5 min before stopping at 0% B, 5 min. The MS conditions were sheath gas (Arb) of 50, aux gas (Arb) of 10, sweep gas (Arb) of 1, ion source voltage of 3500 V, H-ESI ionization source, vaporizer temperature of 350 °C, cycle time 0.4 s and collision gas pressure of 2 mTorr. The MRM transitions of precursors and product ions were: NAD$^+$ precursor m/z 664.291/product ions m/z 428.137, 542.220; $^{13}$C-NAD$^+$ precursor m/z 669.262/product ions m/z = 428.196, 547.220; NADP$^+$ precursor m/z 744.083/product ions m/z 508.071, 603.982 and 622.125. Raw files were imported into Xcalibur 3.1 and Freestyle (Thermo Fisher Scientific) for quantification. Peaks were selected according to the retention time and product ions of the commercially available standard compounds of NAD$^+$, $^{13}$C-NAD$^+$ and NADP$^+$. For quantification of NAD$^+$, $^{13}$C-NAD$^+$ and NADP$^+$, standard curves were generated with of 50, 75, 100, 300, 500, 750, 1000, 2000, 3000, 4000 and 5000 pg on column. The data of three biological samples with three technical replicates for the infiltrated and systemic leaves were calculated.

## Quantification and statistical analysis

The disease resistance results presented in this study are derived from one experimental dataset. These results were confirmed in at least three independent experiments conducted at different times. Both *Psm* and *Psm-lux* were used for disease resistance tests with the CFU and bioluminescence quantification methods, respectively. For the CFU method, eight leaf replicates from eight plants (one leaf from each plant) were collected for bacterial titer assessment. For the bioluminescence method, eight to 20 leaf replicates were used (one leaf from one plant). The eNAD(P) results presented in this study are from measurements of three independent biological samples. Each biological sample consisted of 12 leaves from 12 plants. The NAD(P) and eNAD(P) measurement experiments were independently performed twice or three times. The gene expression results are from analyses of three independent biological samples and each sample was taken from six leaves on six plants (one leaf from each plant). The gene expression experiments were also independently performed twice. Statistical analyses were performed using the data analysis tools (Student's *t* test) in Microsoft Excel of Microsoft Office 2023 for Macintosh as well as one-way ANOVA and two-way ANOVA in Prism 9.

**Reporting summary**

Further information on research design is available in the Nature Portfolio Reporting Summary linked to this article.

## Data availability

The authors declare that all data supporting the findings of this study are available within the manuscript and its supplementary files or are available from the corresponding author upon request. RNA-seq data generated as part of this study has been deposited in the Gene Expression Omnibus repository under accession code GSE225107. Source data are provided with this paper.

## Code availability

The scripts developed in house at UF ICBR are available on request.

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

## Acknowledgements

We thank Jeongim Kim for sharing the pOpON vector with us. We thank Akira Mine for sharing with us the step-by-step protocol for generation of the *luxCDABE*-tagged *Psm* (*Psm _lux*). We thank Jeff Rollins, Jeongim Kim, and Fiona M. Harris for their comments on this manuscript. M.Z. is partially supported by a scholarship from the University of Florida Plant Molecular and Cellular Biology Program. Q.L. was supported by grants from NSF and USDA. M.Z. and Z.M. were partially supported by grants from NSF and USDA.

## Author contributions

Q.L. and Z.M. conceived the study. Q.L. and M.Z. generated the transgenic lines and conducted the pathogen growth, gene expression, NAD(P) measurement, as well as GUS and DAB staining experiments. S.Chhajed and S.Chen conducted the MS/MS experiment. Y.Z. conducted the RNA-seq experiment. F.Y. conducted RNA-seq data analysis. Q.L. and Z. M. wrote the paper with input from all authors.

## Competing interests

The authors declare no competing interests.
