## [Peer Review File · Nature Communications]

N-hydroxypipicolinic acid triggers systemic acquired resistance through extracellular NAD(P)REVIEWER COMMENTS

Reviewer #1 (Remarks to the Author):

1. What are the noteworthy results?

The authors have done many complex and useful experiments that bring together many of the key players in SAR (NHP, eNADP, ROS, LecRK-VI.2) to support their model that SAR mobile signals converge on eNADP in systemic tissues to trigger SAR.

They used the Dex-inducible system to limit expression of the LecRK-VI.2 receptor to lower or upper leaves to pin point the role of this receptor to systemic leaves during SAR. They successfully limited the production of NAD(P) to one leaf using the Dex-inducible FIN4/fin4 transgenic line, allowing them to show that NAD(P) actually moves to systemic leaves. This is the only way to prove that a signal, in this case a SAR signal actually moves to distant tissues. To my knowledge, this has only been done for one other SAR long-distance signal – DIR1

2. Will the work be of significance to the field and related fields? How does it compare to the established literature? If the work is not original, please provide relevant references. This work is original and very useful for the SAR mobile signaling field and for those that study local resistance responses (PTI, ETI).

3. Does the work support the conclusions and claims, or is additional evidence needed? For the most part yes, see my comments below.

4. Are there any flaws in the data analysis, interpretation and conclusions? Do these prohibit publication or require revision?

For the most part the experiments are well done, there are some conclusions that are too strong/definitive (see comments below)

5. Is the methodology sound? Does the work meet the expected standards in your field? For the most part – see comments below

6. Is there enough detail provided in the methods for the work to be reproduced?

Some clarifications are needed

- The number of times each experiment was replicated, is not indicated in the methods or the figure legends, please add this to each figure legend. Plant-microbe experiments (but not the RNA-Seq experiment), must be repeated a number of times (at least 2) and similar results must be observed each time. This is necessary as environmental conditions can

change even in growth chambers and there can be differences in watering from person to person, all of which can affect the plant immune response and bacterial multiplication.

- In Figure 1b and many other figures, Total NAD, NADP or extracellular NADP levels are shown relative to levels in wild type which were arbitrarily set to 1. Please provide a rationale for why this was done and the actual levels in at least some experiments should be shown.

Comments/Questions

Abstract

Line 28 – need to remove eNAD(P) and its receptor, the lectin receptor kinase (LecRK), as to my knowledge this has not been proven, but rather past papers suggest this

Line 30 – change to ... (NHP), are thought to move to systemic....

Line 31 – change leakage to accumulation here and many other places in the manuscript, because leakage was not measured in this paper.

Line 34, change demonstrate to suggest

Introduction

Line 53 – also talk about and cite the paper that demonstrated that DIR1 moves to distant leaves during SAR (Champigny et al, FPS 2013, vol3, article 230) as this is the only paper to limit a SAR signal to 1 leaf, to show that DIR1 moves to distant leaves, as has now been shown for eNAD(P) in this manuscript

Line 61 – need to introduce here or later on page 6, where NAD(P) is produced and by what enzyme (FIN4 encodes ???) and then talk about how its thought to get to the intercellular space in leaves.

Line 74 – need to introduce the authors previous paper on eNAD(P)-LecRK/BAK1 (Nat Comm 2019 10:4810), here so that the reader will understand this manuscript

Results

Line 79 – briefly describe the fin4-3 mutant unless it's been done above on Line 61

Fig 1 – Anova lettering combined with brackets in Figure 1d-f and Figure 7b is really not clear. For example, in Fig. 1d, it looks as if SAR- and SAR+ bars have the same letter – a, when clearly they are significantly different (Fig 1d). I recommend use standard Anova letter labeling as in Figure 1b. Or this labeling has to be clearly explained in the figure legends of Figures 1 and 7.

Line 105 - total NAD(P) was determined in leaves as indicated in fig 2, not intracellular NAD(P) – please fix

Line 115 – reference needed for NA converted to NAD inside cells

Line 118 to 119 – change to these increases may be due to the added NAD+...

Lines 124, 125, 143, 145, and many other places in the manuscript – need to say NA treatment or NAD treatment or NHP treatment, etc, not just NA, since you don't know exactly what is happening in the plant after treatment.

Line 124-125 – Basal resistance needs to be defined as it means different things to different people. It would also be clearer if the sentence was changed to – NA treatment restored basal resistance to Psm in fin4-3, whereas only NAD+ treatment induced resistance....

Line 126-127 – Because total NAD(P)+ levels were measured in this experiment and because these data are not definitive, please change to ...these results suggest that NAD(P)+ is immunogenic when present in the extracellular space

Figure 2 – How does the RLU method of determining bacterial levels relate to the standard method of isolating bacteria from leaf discs? This needs to be added or if this has been discussed before in other papers, this needs to be cited. The field needs to be able to compare these RLU/ld data to the many experiments done using the standard bacterial extraction method. Additionally, please write out RLU in the figure legend as this is the first time it is mentioned in the body of the paper.

Line 131 – please write out extracellular NAD(P), since this is the first time you discuss this in the Results and briefly describe how it is measured. Briefly remind the reader why you are looking at LecRK-VI.2.

Line 150 change to ...and suggests that eNAD(P)....

Line 153 – change to NAD(P) produced at the....

Line 179, change to morphology (Fig. 4g)....

Line 180, change to type plants (Supplementary Fig. 7)

Line 186-187, change to.. into the lower leaves followed by SAR induction 24 hours later, this triggered accumulation of significantly...

Line 189, change to SAR induction results in NAD(P) production in lower leaves to form eNAD(P)...

Line 189 – it's not clear what you mean by to form eNAD(P) and this was not determined in this experiment. It would be beneficial to talk about how NAD(P) is thought to get to the

extracellular space.

Figure 4 legend and other legends where AWFs were collected

– it is necessary to indicate that Apoplastic Washing Fluids (AWFs) were collected from leaves given that this figure is titled eNAD(P) is a mobile SAR signal

- Fig. 4 title is too definitive and so is line 193-194, as this data shows that eNAD(P) moves to distant tissues, so it is mobile and may be a SAR mobile signal, but to prove it's a SAR mobile signal, one has to figure out what it is doing in distant tissues. This has not been shown in this paper, although the authors have provided compelling evidence that supports the idea that eNAD(P) may interact with the LecRK receptor in distant leaves as part of the establishment/priming stage of SAR.

Fig. 4, line 446, change to...The upper systemic leaf was collected 48 hr later and AWFs were then extracted.

Line 200, change to... when FIN4 was expressed in Dex-treated upper...

Line 203, change indicates to suggests

Line 204 & title of Fig. 5

- not sure what is meant by de novo NAD(P) leakage, as this was measured, but instead extracellular NAD(P) was measured by collecting AWFs from leaves or total NAD(P) was measured by grinding up leaves.

- this needs to be modified everywhere it occurs in the manuscript

Line 206-207 – change to ...indicating that Dex-inducible NAD(P) production in systemic....

Fig. 5, line 459-460 ... biological induction of SAR was conducted – not clear, please fix

Fig.5 line 460, change to, AWFs from the upper systemic leaves were collected 48 hr later

Fig.5, line 466, change to... Twenty-four hours later, biological SAR was induced.

Line 211 – no one has actually shown this, as an experiment in which NHP production is limited to 1 leaf has not been done, to my knowledge, so please fix this sentence.

Lines 215 and 221 – NHP movement was not examined as NHP production was not limited to one leaf, rather NHP was infiltrated into leaves and many molecules can access the phloem for movement to distant leaves when exogenously applied to leaves, so this does not examine NHP movement during a natural SAR response. Additionally, just because a molecule accumulates in the phloem, for example, SA, this does not prove it's a mobile SAR signal. Please modify these sentences to reflect this.

Line 216 – change leakage to accumulation

Line 220 – change leakage to accumulation

Line 224, given that some eNAD(P) accumulated in systemic leaves of fmo1, change to ...NHP not only induces, but also contributes to systemic eNAD(P) accumulation.

Lines 231-232 – don't understand the rationale for why a lower NHP concentration was used – please clarify

Line 236-338, change to – Except for PR1 in lecrk-VI.2, NHP-induced expression of PR1, ALD1 and FMO1 as well as NHP-induced resistance to Psm....

Fig. 6, line 470, change release to accumulation

Fig. 6, line 475 change to - collected and then AWFs were extracted at the indicated times

Fig. 6 – it would make it easier for the reader if the x axis label in a,b,c,d was changed to- hr after NHP treatment in lower leaves – could make 1 label that extend across all 4 figures.

The same is true for Fig 6e,f.

Line 241, change to NHP triggers NAD(P) accumulation through ROS accumulation

Line 243, change to ROS have been...

Line 244, change to leading to pore formation.

Line 244, change to – Methyl viologen (MV) treatment

Line 251, 252, add treatment after NAD(P)+

Line 253, change demonstrate to suggest

What is the significance of indicating NAD(P) or NAD(P)+, are these interchangeable?

Examples on lines 241 and 252

Fig 8 – change title to NHP treatment triggers NAD(P) accumulation through RBOHF-generated ROS

Discussion

Line 271 – Change to ... our data provides compelling support for the idea that systemic eNAD(P) originates from two sources...

Line 272 – change leakage to accumulation

Line 272 – to prove that eNAD(P) is a mobile SAR signal, must not only show it moves to distant tissues during SAR, must demonstrate its function in distant leaves. The data in this paper provides support for the idea that it function by interacting with the LecRK receptor and may be a hub/converging point for other SAR mobile signals

Line 273, change indicate to suggest

Line 273-274 – not really sure what you mean by certain SAR mobile signals converge on

eNAD(P).... This idea needs to be explained in more detail

Line 275, change to – leaves induced extracellular NAD(P) accumulation in...

Line 277, change indicate to suggest

Line 278, change to for triggering extracellular NAD(P) accumulation in...

Line 281, change demonstrate to strongly suggest that....

Line 282-283, change to ... triggering NAD(P) accumulation in the extracellular space in systemic tissues.

Line 284, change to...NHP triggers extracellular NAD(P) accumulation through ROS production.

Line 287, change to ...ROS can induce ion leakage in paraquat-damaged maize leaves,...

Line 288, change exhibit to exhibited

Line 289-290, change to...NHP treatment results in ROS.... and ROS accumulation triggers extracellular NAD(P) accumulation....NHP-induced extracellular NAD(P) accumulation...

Line 292, change induce to induced, change leakage to accumulation

Line 293, change from investigations to investigation

Line 299 – change compromises to compromised

Line 306 & 308-309, change to eNAD(P) accumulation...

Line 311, change to...evidence supports the idea that

Line 312, modify this sentence to - We have shown that NAD(P)+ treatment induces wild-type levels of immunity in dir1-1... (or dir1-2, which mutant allele was used?). Is the reference for this data in reference 16 – it is not clear, please fix.

Line 317 – change indicating to suggesting

Line 326 – References required for this information, especially that DTI induces SAR

Line 333, change to... Based on our findings, we suggest that SAR is ...

Line 340 – change wholistic to holistic

It is recommended that the authors discuss how they think extracellular NAD(P) moves to distant leaves, given that many papers provide evidence that SAR mobile signals move cell-to-cell and then access the phloem for movement to distant leaves and also take into account what is currently known about how molecules or proteins enter the phloem - via plasmodesmata from companion cells to sieve elements. To my knowledge, there is currently no evidence that molecules or proteins access the phloem from the extracellular space.

Reviewer #2 (Remarks to the Author):

The manuscript of Li, Mou and colleagues builds up on a previous study published by the same research group in which the SAR-inducing capacity of exogenously applied NAD(P) in plants and the requirement of the lectin receptor kinase LecR-VI.2 and the co-receptors BAK1/BKK1 as downstream components were identified (Wang et al., Nat Commun 2019). In the current study, they investigate the role of endogenously produced, extracellular NAD(P) [eNAD(P)] in the biological SAR process and how eNAD(P) functions in SAR signal transduction. It is shown that the putative eNAD(P) receptor LecR-VI.2 functions in the systemic tissue in SAR development. Further, the authors suggest that eNAD(P) can act as a mobile signal that moves from the inoculated to the systemic tissue during SAR induction. Interestingly, Li et al. provide insights into the interplay between the recently discovered SAR-inducing metabolite NHP and eNAD(P). This also includes the downstream components of eNAD(P) - LecRK-VI.2/BAK1. Altogether, the findings show that eNAD(P)-LecRK-VI.2/BAK1 act downstream of NHP in SAR induction. The data indicate that NHP induces the release of eNAD(P), which in turn accumulates in the systemic leaves, and that ROS participate in the release of eNAD(P). I think that this study has the potential of providing fundamental new insights into the molecular events underlying SAR in plants. At this stage, I feel that a few additional experiments could improve the manuscript and further validate some the conclusions that are not fully supported by the current data. Further, some of results might be more carefully interpreted, and the corresponding conclusions and final model modified so that they more closely reflect the results.

Specific comments:

1) A central hypothesis of the manuscript is that eNAD(P) travels through the plant during biological SAR (“NAD(P) released at the primary infection site moves systemically”). However, although ¹³C-labelled exogenously added NAD⁺ is shown to move to systemic tissue, a direct movement of endogenously produced eNAD(P) is not shown (which is indeed difficult to accomplish experimentally). This should be made clear in the discussion. I agree, however, that the sophisticated analyses of the DEX-lines (DEX:FIN4-fin4 and DEX:LecRK-lecrk) are consistent with the movement-hypothesis. For example, it is demonstrated that biological SAR induction by bacterial inoculation of local leaves results in the FIN4-

dependent accumulation of eNAD(P) in systemic leaves (Fig. 4). I think that the mobility hypothesis would be further supported by providing data on the local accumulation of eNAD(P): Does eNAD(P) accumulate in the pathogen-inoculated leaves, and does this occur in a FIN4-dependent manner? These data could be added to Fig. 4.

2) In the same context: It is shown that exogenously added NHP results in local and systemic accumulation of eNAD(P) (Fig. 6a-d), and that the systemic accumulation of eNAD(P) upon local induction of biological SAR is attenuated in the NHP biosynthesis mutant *fmo1*. The question that arises here: if eNAD(P) accumulates locally upon bacterial inoculation, is this local accumulation FMO1-dependent? Such a finding would show that NHP induces eNAD(P) release in the local tissue under biological SAR conditions, corroborate the conclusions drawn from the exogenous NHP-data (Fig. 6a-d), and strengthen the movement-hypothesis.

3) Determination of extracellular NAD(P) levels: It would be helpful to be informed in the results section directly that eNAD(P) levels were assessed by analyzing apoplastic washing fluids of leaves. Moreover, the (e)NAD(P) levels are always given as relative levels which unfortunately does not provide any information on the actual, absolute quantities of NAD(P) or eNAD(P) in the leaf tissue. The absolute levels are of great interest, however, and provide information about the physiological levels of (e)NAD(P) involved in SAR induction. This could then be directly compared with the exogenous (millimolar) amounts of NAD(P) used to induce SAR chemically (Fig. 2f).

4) The SAR resistance assays with *lerck-IV* and *bak1/bkk1* mutants in the current study and the previous, Wang et al. (2019) study indicate attenuated SAR in these lines; however, a small SAR response seems to be present in these lines robustly (including a residual transcriptional SAR response in *lecrk*). This indicates the existence of another downstream signaling component independent of these genes. The same is seen for the NHP-triggered SAR (Fig. 7): a residual, NHP-triggered but *lecrk-VI.2/bak1/bkk1*-independent pathway to SAR. This should be discussed and illustrated in the final model (see below). Even in *fin4*, a small NHP-response seems to exist (Fig. 4). Could you explain this? Might there be a small eNAD(P)-independent pathway to SAR?

5) Regarding rboh/ROS-dependency of NHP-induced SAR and eNAD release: The authors state that “NHP induces systemic eNAD(P) accumulation and systemic immunity through RBOH-produced ROS”. On the basis of the presented data, this is an overinterpretation. I agree that the NHP-induced responses are attenuated in the rboh lines, but there are still considerable NHP-inducible resistance- and eNAD(P)-responses in these lines (Fig. 8). So, the statements in the manuscript throughout (including abstract and final model) on the key role of ROS should be toned down to some extent (e.g. “Rboh-dependent ROS formation positively affects but is not fully required for the NHP-triggered NAD(P) leakage and SAR responses).

6) The ANOVA of the treatment-related (SAR, NHP) resistance data tests for differences in the induction of resistance in different lines but unfortunately does not test whether differences between treatment samples and control samples exist within individual lines (e.g. Fig. 1d,f; Fig. 7a). However, I think that this information is of interest (see my comments 4 and 5). Therefore, I suggest to compare for both the treatment-related and the genotypic differences in all of these data sets by ANOVA (as performed for eNADP levels in Fig. 8h, for example).

7) In my opinion, the final model Fig. 8k) should be improved by focussing on the actually, experimentally obtained evidence about eNADP, NHP and ROS in this and the Wang et al. (2019) study: A) in the local, 1° tissue, an arrow from NHP to eNAD(P) is missing (see results on NHP-induced local eNADP accumulation, Fig. 6a,b). Otherwise, an NHP-independent but eNAD-dependent signaling pathway is implicated, which is not the case (NHP accumulation is indispensable for SAR; NHP-deficient *ald1* and *fmo1* mutants are fully compromised in SAR). B) The depicted roles of G3P, AzA, NO, and volatiles in NHP-induced SAR are hypothetical and not experimentally verified, so I suggest to remove them from the model. C) As stated above, the function of ROS seems over-stated because the model indicates incorrectly that all the NHP-triggered SAR signaling would proceed via ROS. D) In my opinion, SA and NPR1, two key SAR components, are missing in the model. It was shown in the Wang et al. (2019)-study that eNADP-triggered SAR requires NPR1 and strongly depends on *sid2*, so SA and NPR1 act downstream of eNAD(P) [and also downstream of NHP – see Hartmann et al., Cell 2018; Liu et al., Plant Cell 2020; Yildiz et al. Plant Physiol 2021). Thus, it

would make sense to place SA and NPR1 downstream of eNAD(P) in the model. E) A modest NHP-dependent but *lcrk-VI.2/bak1/bkk1*-independent pathway to SAR seems to exist (see comment 4).

8) Discussion: the above-mentioned downstream function of SA and NPR1 in NHP- and eNAD(P)-mediated SAR are not properly discussed in my opinion. Also, it is worth considering that TGA transcription factors function as key downstream components of the NHP-triggered (transcriptional) SAR response (Yildiz et al., PCE 2023).

9) Introduction, line 39: "Plants lack ... adaptive immunity ...": Doesn't SAR represent a form of adaptive immunity?

Responses to the Reviewers' comments

We would like to thank both reviewers for the comments and suggestions, which helped us to significantly improve the manuscript.

Reviewer #1:

1. What are the noteworthy results?

The authors have done many complex and useful experiments that bring together many of the key players in SAR (NHP, eNADP, ROS, LecRK-VI.2) to support their model that SAR mobile signals converge on eNADP in systemic tissues to trigger SAR.

They used the Dex-inducible system to limit expression of the LecRK-VI.2 receptor to lower or upper leaves to pin point the role of this receptor to systemic leaves during SAR. They successfully limited the production of NAD(P) to one leaf using the Dex-inducible FIN4/fin4 transgenic line, allowing them to show that NAD(P) actually moves to systemic leaves. This is the only way to prove that a signal, in this case a SAR signal actually moves to distant tissues. To my knowledge, this has only been done for one other SAR long-distance signal – DIR1

2. Will the work be of significance to the field and related fields? How does it compare to the established literature? If the work is not original, please provide relevant references.

This work is original and very useful for the SAR mobile signaling field and for those that study local resistance responses (PTI, ETI).

3. Does the work support the conclusions and claims, or is additional evidence needed?

For the most part yes, see my comments below.

4. Are there any flaws in the data analysis, interpretation and conclusions? Do these prohibit publication or require revision?

For the most part the experiments are well done, there are some conclusions that are too strong/definitive (see comments below)

5. Is the methodology sound? Does the work meet the expected standards in your field?

For the most part – see comments below

6. Is there enough detail provided in the methods for the work to be reproduced?

Some clarifications are needed

A: We thank the reviewer for the very encouraging comments.

- The number of times each experiment was replicated, is not indicated in the methods or the figure legends, please add this to each figure legend. Plant-microbe experiments (but not the RNA-Seq experiment), must be repeated a number of times (at least 2) and similar results must be observed each time. This is necessary as environmental conditions can change even in growth chambers and there can be differences in watering from person to person, all of which can affect the plant immune response and bacterial multiplication.

A: We described the number of times each experiment was replicated in the Methods under Quantification and statistical analysis at line 824 to line 829 in the original manuscript: “The disease resistance results presented in this study are derived from one experimental dataset. These results were confirmed in at least three independent experiments conducted at different times. Both *Psm* and *Psm-lux* were used for disease resistance tests with the CFU and bioluminescence quantification methods, respectively. For the CFU method, eight leaf replicates from eight plants (one leaf from each plant) were collected for bacterial titer assessment. For the bioluminescence method, eight to 20 leaf replicates were used (one leaf from one plant).” We added the information in each figure legend.

- In Figure 1b and many other figures, Total NAD, NADP or extracellular NADP levels are shown relative to levels in wild type which were arbitrarily set to 1. Please provide a rationale for why this was done and the actual levels in at least some experiments should be shown.

A: We use the Promega NAD(P)/NAD(P)H-Glo Assay kits to measure NAD and NADP. This method is based on enzymatic cycling assays. The bioluminescent intensities vary significantly if the kit is used at different times. This is probably because the enzyme activity is reduced if the kit is kept in freezer for too long. We also noticed differences among different batches of the kits ordered at different times. The NAD(P) measurement experiments described in the manuscript were conducted in the past 2-3 years. The relative light unit (RLU) values vary significantly. Since we were interested in the differences between genotypes/treatments (and the reagents are also expensive), we did not make standard curves when we conducted the experiments. Therefore, we arbitrarily set the levels in wild type or mock-treated wild type to 1 to allow comparison across different experiments. A brief rationale “allowing comparison across experiments” was added in Fig. 1b, c (total NAD(P)) and Fig. 4a, b (eNAD(P)) legends.

The most important question is whether SAR induction leads to systemic eNAD(P) accumulation to levels that are reached upon exogenous NAD(P)⁺ treatment. To determine the levels of eNAD(P) in systemic leaves during SAR induction and compare the levels with those upon exogenous NAD(P)⁺ application, we infiltrated three lower leaves with 1 mM MgCl₂, *Psm*, water, 0.5 mM NAD⁺, or 1 mM NAD⁺. Upper systemic leaves were collected and weighed for apoplastic washing fluid (AWF) collection. For MgCl₂ and *Psm*, the systemic leaves were collected at 48 hours after the infiltration, and for water, 0.5 mM NAD⁺, and 1 mM NAD⁺, the systemic leaves were collected at 4 hours after the infiltration. After measurement, the eNAD levels were calculated to µg/g fresh weight (FW) (it is difficult to calculate molar concentration as the volume of actual apoplastic fluids is difficult to define). The eNAD levels in the systemic leaves at 48 hours after local inoculation with *Psm* is in the range of 30-40 µg/g FW, which is similar to those in the systemic leaves at 4 hours after local infiltration of 0.5 mM NAD⁺.

We added this result in the text as “To compare systemic eNAD(P) accumulation during SAR induction and exogenous NAD(P)⁺ treatment, we infiltrated lower leaves with 1 mM MgCl₂, *Psm*, water, 0.5 mM NAD⁺, or 1 mM NAD⁺. Systemic leaves were collected four hours after the water and NAD⁺ treatments and 48 hours after the MgCl₂ and *Psm* treatments for eNAD assays. The systemic eNAD levels upon SAR induction was in the range of 30-40 µg/g fresh weight (FW), which is comparable to those after 0.5 mM NAD⁺ treatment (Supplementary Fig. 6).” We hope that this result could address the reviewer’s concern.

Comments/Questions

Abstract

Line 28 – need to remove eNAD(P) and its receptor, the lectin receptor kinase (LecRK), as to my knowledge this has not been proven, but rather past papers suggest this

A: We agree with the reviewer that past papers suggest (not proved) that LecRK-VI.2 is an eNAD(P) receptor. We removed “its receptor” but not “eNAD(P)”, as eNAD(P) is also required in systemic leaves.

Line 30 – change to ... (NHP), are thought to move to systemic....

A: We changed to “putative mobile signals, e.g., N-hydroxypipicolinic acid (NHP), trigger de novo systemic eNAD(P) accumulation ---”.

Line 31 – change leakage to accumulation here and many other places in the manuscript, because leakage was not measured in this paper.

A: As suggested, “leakage” has been changed to “accumulation” here and throughout the manuscript where appropriate.

Line 34, change demonstrate to suggest

A: As suggested, “demonstrate” has been changed to “suggest”.

Introduction

Line 53 – also talk about and cite the paper that demonstrated that DIR1 moves to distant leaves during SAR (Champigny et al, FPS 2013, vol3, article 230) as this is the only paper to limit a SAR signal to 1 leaf, to show that DIR1 moves to distant leaves, as has now been shown for eNAD(P) in this manuscript

A: We thank the reviewer for providing this information. As suggested, the sentence “**Among these signals, DIR1 has been demonstrated to move down the leaf petiole to distant leaves upon SAR induction**” has been added and the reference has been cited.

Line 61 – need to introduce here or later on page 6, where NAD(P) is produced and by what enzyme (FIN4 encodes ???) and then talk about how its thought to get to the intercellular space in leaves.

A: This background information has been added at the beginning of Results before the *fin4-3* mutant is mentioned: “**Since NAD(P) is believed to leak into the extracellular space upon pathogen-caused cell damage, we tested whether cellular NAD(P) plays a role in SAR. To this end, we took advantage of the previously reported NAD biosynthesis mutant *flagellin-insensitive4-3 (fin4-3)* that carries a T-DNA insertion toward the 3’ end of the *FIN4* gene. *FIN4***”

encodes the chloroplastic enzyme aspartate oxidase that catalyzes the first irreversible step in de novo NAD biosynthetic pathway.”

Line 74 – need to introduce the authors previous paper on eNAD(P)-LecRK/BAK1 (Nat Comm 2019 10:4810), here so that the reader will understand this manuscript

A: We thank the reviewer for this suggestion. The background information “We have previously reported that exogenously added NAD⁺ moves systemically and that the lectin receptor kinase (LecRK), LecRK-VI.2, is a putative NAD(P) receptor and plays a pivotal role in biological induction of SAR. We have also shown that BRASSINOSTEROID INSENSITIVE1-ASSOCIATED KINASE1 (BAK1) constitutively associates with LecRK-VI.2 and functions in eNAD(P) signaling and SAR. Although these results suggest that eNAD(P) is a potential SAR mobile signal,” has been added.

Results

Line 79 – briefly describe the fin4-3 mutant unless it’s been done above on Line 61

A: As the reviewer suggested, we added the related background information at this place: “Since NAD(P) is believed to leak into the extracellular space upon pathogen-caused cell damage, we tested whether cellular NAD(P) plays a role in SAR. To this end, we took advantage of the previously reported NAD biosynthesis mutant *flagellin-insensitive4-3 (fin4-3)* that carries a T-DNA insertion toward the 3’ end of the *FIN4* gene. *FIN4* encodes the chloroplastic enzyme aspartate oxidase that catalyzes the first irreversible step in the de novo NAD biosynthetic pathway.”

Fig 1 – Anova lettering combined with brackets in Figure 1d-f and Figure 7b is really not clear. For example, in Fig. 1d, it looks as if SAR- and SAR+ bars have the same letter – a, when clearly they are significantly different (Fig 1d). I recommend use standard Anova letter labeling as in Figure 1b. Or this labeling has to be clearly explained in the figure legends of Figures 1 and 7.

A: We apologize for the confusion. To make the figures clear, we have changed two-way ANOVA analyses in Fig. 1d,f; Fig. 7b,c; Fig. 8i to one-way ANOVA analyses. We kept two-way ANOVA analysis for Fig. 7a, since one-way ANOVA does not reveal the effects of genotypes on the treatment in this case. The differences between treatment and control in Fig. 1d,f; Fig. 7b,c; Fig. 8i are discussed in the manuscript. “The 35S:CD38 transgene significantly inhibited NHP-induced systemic immunity.” has been added in Fig. 7a legends.

Line 105 - total NAD(P) was determined in leaves as indicated in fig 2, not intracellular NAD(P) – please fix

A: As the reviewer pointed out, “intracellular” has been changed to “total”.

Line 115 – reference needed for NA converted to NAD inside cells

A: A reference about NA converted to NAD inside cells has been added.

Line 118 to 119 – change to these increases may be due to the added NAD+...

A: As the reviewer suggested, “were” has been changed to “may be”.

Lines 124, 125, 143, 145, and many other places in the manuscript – need to say NA treatment or NAD treatment or NHP treatment, etc, not just NA, since you don’t know exactly what is happening in the plant after treatment.

A: As the reviewer suggested, “treatment” has been added in these and other places throughout the manuscript.

Line 124-125 – Basal resistance needs to be defined as it means different things to different people. It would also be clearer if the sentence was changed to – NA treatment restored basal resistance to Psm in fin4-3, whereas only NAD+ treatment induced resistance....

A: We thank the reviewer for this suggestion. The sentence has been changed to “**NA treatment restored basal resistance to Psm (the resistance activated by Psm on the wild-type plants) in fin4-3, whereas only NAD⁺ treatment induced resistance to Psm in wild type and fin4-3.**”

Line 126-127 – Because total NAPD(P)+ levels were measured in this experiment and because these data are not definitive, please change to ...these results suggest that NAP(P)+ is immunogenic when present in the extracellular space

A: As the reviewer suggested, “demonstrate” has been changed to “suggest”.

Figure 2 – How does the RLU method of determining bacterial levels relate to the standard method of isolating bacteria from leaf discs? This needs to be added or if this has been discussed before in other papers, this needs to be cited. The field needs to be able to compare these RLU/ld data to the many experiments done using the standard bacterial extraction method Additionally, please write out RLU in the figure legend as this is the first time it is mentioned in the body of the paper.

A: A detailed comparison between the RLU method and the traditional method has been made by Matsumoto et al. (2022, Plant Commun; the original reference 52). We added this reference in Methods under quantification of Psm growth. “**RLU: relative light unit.**” has been added in Fig. 2e,f legends.

Line 131 – please write out extracellular NAD(P), since this is the first time you discuss this in the Results and briefly describe how it is measured. Briefly remind the reader why you are looking at LecRK-VI.2.

A: A sentence “**We have previously shown that the lectin receptor kinase (LecRK), LecRK-VI.2, is a potential receptor for extracellular NAD(P) [eNAD(P)].**” has been added at the beginning of the paragraph to introduce LecRK-VI.2. How eNAD(P) is measured was briefly described by

adding “Note that eNAD(P) levels were assessed by analyzing apoplastic washing fluids (AWFs) of the indicated leaves.” when the first eNAD(P) measurement was described (Fig. 4a,b).

Line 150 change to ...and suggests that eNAD(P)....

A: As the reviewer suggested, “indicate” has been changed to “suggest”.

Line 153 – change to NAD(P) produced at the....

A: This was changed to “eNAD(P) accumulated at the primary infection ---”. It is not NAD(P) produced in the local leaves.

Line 179, change to morphology (Fig. 4g)....

A: Change has been made as suggested.

Line 180, change to type plants (Supplementary Fig. 7)

A: Change has been made as suggested.

Line 186-187, change to.. into the lower leaves followed by SAR induction 24 hours later, this triggered accumulation of significantly...

A: We thank the reviewer for this suggestion. Changes have been made as suggested.

Line 189, change to SAR induction results in NAD(P) production in lower leaves to form eNAD(P)...

Line 189 – it’s not clear what you mean by to form eNAD(P) and this was not determined in this experiment. It would be beneficial to talk about how NAD(P) is thought to get to the extracellular space.

A: We thank the reviewer for pointing out this confusion. We added a sentence “It has previously been shown that pathogen infection leads to eNAD(P) accumulation in the inoculated leaves, which is likely due to pathogen-caused cell damage.”, and changed “These results indicate that SAR induction causes NAD(P) release in the lower leaves to form eNAD(P),” to “These results indicate that SAR induction results in accumulation of eNAD(P) in the lower leaves,”. We hope this has clarified the confusion.

Figure 4 legend and other legends where AWFs were collected

– it is necessary to indicate that Apoplastic Washing Fluids (AWFs) were collected from leaves given that this figure is titled eNAD(P) is a mobile SAR signal

A: As the reviewer suggested, we added “Note that eNAD(P) levels were assessed by analyzing apoplastic washing fluids (AWFs) of the indicated leaves.” when the first eNAD(P) measurement was described in the text and in Fig. 4a, b legends as well. It is also described in Methods under eNAD and eNADP measurement.

- Fig. 4 title is too definitive and so is line 193-194, as this data shows that eNAD(P) moves to distant tissues, so it is mobile and may be a SAR mobile signal, but to prove it's a SAR mobile signal, one has to figure out what it is doing in distant tissues. This has not been shown in this paper, although the authors have provided compelling evidence that supports the idea that eNAD(P) may interact with the LecRK receptor in distant leaves as part of the establishment/priming stage of SAR.

A: We agree with the reviewer that it is too soon to definitively claim that eNADP is a mobile SAR signal. The title of Fig. 4 has been changed to “eNAD(P) is a potential mobile SAR signal”. We also changed “demonstrate” to “suggest” in the text to reflect this possibility.

Fig. 4, line 446, change to...The upper systemic leaf was collected 48 hr later and AWFs were then extracted.

A: Changes have been made as the reviewer suggested.

Line 200, change to... when FIN4 was expressed in Dex-treated upper...

A: Changes have been made as suggested.

Line 203, change indicates to suggests

A: As the reviewer suggested, “indicates” has been changed to “suggests”.

Line 204 & title of Fig. 5

- not sure what is meant by de novo NAD(P) leakage, as this was measured, but instead extracellular NAD(P) was measured by collecting AWFs from leaves or total NAD(P) was measured by grinding up leaves.

- this needs to be modified everywhere it occurs in the manuscript

A: We thank the reviewer for pointing out this confusion. As the reviewer indicated, we measured eNAD(P) by collecting AWFs from the leaves. Thus, we have changed “de novo NAD(P) leakage” to “de novo eNAD(P) accumulation” throughout the manuscript. We hope this change makes sense to the reviewer and future readers.

Line 206-207 – change to ...indicating that Dex-inducible NAD(P) production in systemic....

A: We sincerely apologize for the confusion. It is not “Dex-inducible NAD(P) production in systemic leaves” that plays a central role in SAR. Dex treatment restored NAD(P) levels in the systemic leaves. After SAR induction, mobile signals produced in the lower inoculated leaves moved to the systemic leaves where they triggered eNAD(P) accumulation. It is the eNAD(P) that accumulates in the systemic leaves plays a central role in SAR. We have changed “indicating that de novo NAD(P) leakage in systemic leaves plays a central role in the establishment of SAR.” to “suggesting that de novo eNAD(P) accumulation in systemic leaves

plays a central role in the establishment of SAR.” We hope these changes have clarified the confusion.

Fig. 5, line 459-460 ... biological induction of SAR was conducted – not clear, please fix

A: “biological induction of SAR was conducted” has been changed to “three lower leaves on each plant were infiltrated with 1 mM MgCl₂ (-SAR) or *Psm* (+SAR)”.

Fig.5 line 460, change to, AWFs from the upper systemic leaves were collected 48 hr later

A: Changes have been made as the reviewer suggested.

Fig.5, line 466, change to... Twenty-four hours later, biological SAR was induced.

A: Changes have been made as the reviewer suggested.

Line 211 – no one has actually shown this, as an experiment in which NHP production is limited to 1 leaf has not been done, to my knowledge, so please fix this sentence.

A: The sentence has been changed to “NHP has recently been shown to be a potential SAR mobile signal.”

Lines 215 and 221 – NHP movement was not examined as NHP production was not limited to one leaf, rather NHP was infiltrated into leaves and many molecules can access the phloem for movement to distant leaves when exogenously applied to leaves, so this does not examine NHP movement during a natural SAR response. Additionally, just because a molecule accumulates in the phloem, for example, SA, this does not prove it’s a mobile SAR signal. Please modify these sentences to reflect this.

A: These sentences have been changed to “We then tested if local application of NHP could induce de novo eNAD(P) accumulation in systemic leaves using the *Dex:FIN4/fin4-3* transgenic plants.” and “These results indicate that NHP and/or NHP-induced mobile signals not only induce eNAD(P) accumulation in treated leaves, but also trigger de novo systemic eNAD(P) accumulation.”

Line 216 – change leakage to accumulation

A: Done.

Line 220 – change leakage to accumulation

A: Done.

*Line 224, given that some eNAD(P) accumulated in systemic leaves of *fmo1*, change to ...NHP not only induces, but also contributes to systemic eNAP(P) accumulation.*

A: Based on Reviewer #2's suggestion, we measured eNAD(P) levels in pathogen-infected *fmo1* leaves and found that eNAD(P) levels in the local *Psm*-infected leaves of *fmo1* were not significantly different from those in the wild type. Based on this new result, we changed the sentence to “**Since the *fmo1* mutant is completely SAR defective, these results indicate that NHP-mediated systemic eNAD(P) accumulation plays a crucial role in the establishment of SAR.**”

Lines 231-232 – don't understand the rationale for why a lower NHP concentration was used – please clarify

A: CD38 is not highly active when expressed in the extracellular space in Arabidopsis plants. It can only slightly reduce eNAD(P) levels and partially compromise SAR in *35S:CD38* transgenic plants. When high concentrations (0.5 mM and above) of NHP were used, CD38 had no effect on NHP-induced systemic immunity. This is why we used a lower concentration (0.25 mM) of NHP that would induce less eNAD(P) accumulation. The partial suppression of a low concentration of NHP-induced systemic immunity is consistent with the partial SAR phenotype of the *35S:CD38* plants. We added “**and the *35S:CD38* transgenic plants exhibit partially compromised SAR**” to introduce more background information about the *35S:CD38* transgenic plants. We hope this could clarify the reviewer's concern.

Line 236-338, change to – Except for PR1 in lecrk-VI.2, NHP-induced expression of PR1, ALD1 and FMO1 as well as NHP-induced resistance to Psm....

A: Done.

Fig. 6, line 470, change release to accumulation

A: Done.

Fig. 6, line 475 change to - collected and then AWFs were extracted at the indicated times

A: Was changed to “**collected at the indicated times and then AWFs were collected**”.

Fig. 6 – it would make it easier for the reader if the x axis label in a,b,c,d was changed to- hr after NHP treatment in lower leaves – could make 1 label that extend across all 4 figures. The same is true for Fig 6e,f.

A: We thank the reviewer for this suggestion. We added “in lower leaves” in the x axis label in Fig. 6a, b, c, d and also added “lower leaves” in Fig. 6e, f. We would like to keep Fig. 6a, b, c, d separate as they are individual figures.

Line 241, change to NHP triggers NAD(P) accumulation through ROS accumulation

A: As the reviewer suggested, “NHP triggers NAD(P) leakage through ROS” has been changed to “**NHP triggers eNAD(P) accumulation through ROS**”.

Line 243, change to ROS have been...

A: Done.

Line 244, change to leading to pore formation.

A: Done.

Line 244, change to – Methyl viologen (MV) treatment

A: Done.

Line 251, 252, add treatment after NAD(P)+

A: Done.

Line 253, change demonstrate to suggest

A: Done.

*What is the significance of indicating NAD(P) or NAD(P)+, are these interchangeable?
Examples on lines 241 and 252*

A: NAD(P) includes NAD⁺, NADH, NADP⁺, and NADPH, whereas NAD(P)⁺ includes NAD⁺ and NADP⁺. In our initial work (Zhang et al., 2009, Plant J 57, 302-312), NAD⁺, NADH, NADP⁺, and NADPH were used, and all can induce strong immune responses in Arabidopsis. In this manuscript, only NAD⁺ and NADP⁺ were used for treating plants, whereas NAD (NAD⁺ and NADH) and NADP (NADP⁺ and NADPH) were measured.

Fig 8 – change title to NHP treatment triggers NAD(P) accumulation through RBOHF-generated ROS

A: Changes have been made as suggested.

Discussion

Line 271 – Change to ... our data provides compelling support for the idea that systemic eNAD(P) originates from two sources...

A: Thanks for the suggestion. The sentence has been changed to “our data obtained using the Dex:FIN4/fin4-3 transgenic plants in which NAD biosynthesis can be spatially controlled by Dex application provide compelling support for the idea that systemic eNAD(P) originates from two sources”.

Line 272 – change leakage to accumulation

A: Done.

Line 272 – to prove that eNAD(P) is a mobile SAR signal, must not only show it moves to distant tissues during SAR, must demonstrate its function in distant leaves. The data in this paper provides support for the idea that it function by interacting with the LecRK receptor and may be a hub/converging point for other SAR mobile signals

A: We agree with the reviewer. “as a mobile signal” has been changed to “as a potential mobile signal”.

Line 273, change indicate to suggest

A: Done.

Line 273-274 – not really sure what you mean by certain SAR mobile signals converge on eNAD(P).... This idea needs to be explained in more detail

A: The sentence has been changed to “These results suggest that SAR mobile signals might converge on eNAD(P) in systemic tissues to trigger SAR”. This is based on the result that mobile signals other than eNAD(P) can move to systemic tissues to trigger de novo eNAD(P) accumulation in the systemic tissues.

Line 275, change to – leaves induced extracellular NAD(P) accumulation in...

A: Done.

Line 277, change indicate to suggest

A: Done.

Line 278, change to for triggering extracellular NAD(P) accumulation in...

A: The whole sentence has been re-written to “Since SAR is completely absent in the *fmo1* mutant, these results suggest that eNAD(P) originating from the local leaves is not sufficient for activating effective SAR and that de novo eNAD(P) accumulation in systemic tissues triggered by NHP and possibly other FMO1-dependent mobile signals is essential for SAR establishment.”

Line 281, change demonstrate to strongly suggest that....

A: Done.

Line 282-283, change to ... triggering NAD(P) accumulation in the extracellular space in systemic tissues.

A: As suggested, “triggering NAD(P) leakage” has been changed to “triggering eNAD(P) accumulation”.

Line 284, change to...NHP triggers extracellular NAD(P) accumulation through ROS production.

A: Done.

Line 287, change to ...ROS can induce ion leakage in paraquat-damaged maize leaves,...

A: Done.

Line 288, change exhibit to exhibited

A: Done.

Line 289-290, change to...NHP treatment results in ROS.... and ROS accumulation triggers extracellular NAD(P) accumulation....NHP-induced extracellular NAD(P) accumulation...

A: Done.

Line 292, change induce to induced, change leakage to accumulation

A: Done.

Line 293, change from investigations to investigation

A: Done.

Line 299 – change compromises to compromised

A: Done.

Line 306 & 308-309, change to eNAD(P) accumulation...

A: Done.

Line 311, change to...evidence supports the idea that

A: Done.

*Line 312, modify this sentence to - We have shown that NAD(P)⁺ treatment induces wild-type levels of immunity in *dir1-1*... (or *dir1-2*, which mutant allele was used?). Is the reference for this data in reference 16 – it is not clear, please fix.*

A: Changes has been made as suggested. The *dir1* mutant is *dir1-1* in Ws background that was reported by Maldonado et al. in 2002. NAD(P)⁺ treatment of *dir1* was conducted in the original reference 16 (Wang et al., 2019) Figure S4. This reference (original 16) has been added here.

Line 317 – change indicating to suggesting

A: Done.

Line 326 – References required for this information, especially that DTI induces SAR

A: The PAMP flg22 and the well-studied DAMP Pep1 can induce SAR. The references have been added.

Line 333, change to... Based on our findings, we suggest that SAR is ...

A: Done.

Line 340 – change wholistic to holistic

A: Done.

It is recommended that the authors discuss how they think extracellular NAD(P) moves to distant leaves, given that many papers provide evidence that SAR mobile signals move cell-to-cell and then access the phloem for movement to distant leaves and also take into account what is currently known about how molecules or proteins enter the phloem - via plasmodesmata from companion cells to sieve elements. To my knowledge, there is currently no evidence that molecules or proteins access the phloem from the extracellular space.

A: We thank the reviewer for this recommendation. We changed “movement of eNAD(P) to systemic tissues” to “movement of eNAD(P) through the apoplastic route to systemic tissues” in the first paragraph of Discussion. eNAD(P) must go through the apoplastic route. It cannot be transported into cells including the phloem, because it will no longer be a signal molecule once it enters cells. There are plenty of NAD(P) inside the cell (total ~2 mM), but they do not activate immune responses. Only when NAD(P) is in the extracellular space does it become immune active.

Reviewer #2:

The manuscript of Li, Mou and colleagues builds up on a previous study published by the same research group in which the SAR-inducing capacity of exogenously applied NAD(P) in plants and the requirement of the lectin receptor kinase LecR-VI.2 and the co-receptors BAK1/BKK1 as downstream components were identified (Wang et al., Nat Commun 2019). In the current study, they investigate the role of endogenously produced, extracellular NAD(P) [eNAD(P)] in the biological SAR process and how eNAD(P) functions in SAR signal transduction. It is shown that the putative eNAD(P) receptor LecR-VI.2 functions in the systemic tissue in SAR development. Further, the authors suggest that eNAD(P) can act as a mobile signal that moves from the inoculated to the systemic tissue during SAR induction. Interestingly, Li et al. provide insights into the interplay between the recently discovered SAR-inducing metabolite NHP and eNAD(P).

This also includes the downstream components of eNAD(P) - LecRK-VI.2/BAK1. Altogether, the findings show that eNAD(P)-LecRK-VI.2/BAK1 act downstream of NHP in SAR induction. The data indicate that NHP induces the release of eNAD(P), which in turn accumulates in the systemic leaves, and that ROS participate in the release of eNAD(P). I think that this study has the potential of providing fundamental new insights into the molecular events underlying SAR in plants. At this stage, I feel that a few additional experiments could improve the manuscript and further validate some the conclusions that are not fully supported by the current data. Further, some of results might be more carefully interpreted, and the corresponding conclusions and final model modified so that they more closely reflect the results.

A: We thank the reviewer for the very positive comments.

Specific comments:

1) A central hypothesis of the manuscript is that eNAD(P) travels through the plant during biological SAR (“NAD(P) released at the primary infection site moves systemically”). However, although ¹³C-labelled exogenously added NAD⁺ is shown to move to systemic tissue, a direct movement of endogenously produced eNAD(P) is not shown (which is indeed difficult to accomplish experimentally). This should be made clear in the discussion. I agree, however, that the sophisticated analyses of the DEX-lines (DEX:FIN4-fin4 and DEX:LecRK-lecrk) are consistent with the movement-hypothesis. For example, it is demonstrated that biological SAR induction by bacterial inoculation of local leaves results in the FIN4-dependent accumulation of eNAD(P) in systemic leaves (Fig. 4). I think that the mobility hypothesis would be further supported by providing data on the local accumulation of eNAD(P): Does eNAD(P) accumulate in the pathogen-inoculated leaves, and does this occur in a FIN4-dependent manner? These data could be added to Fig. 4.

A: We agree with the reviewer that it is highly challenging to demonstrate direct movement of the endogenous eNAD(P). Results obtained using the DEX:FIN4/fin4 transgenic lines can only provide support for the hypothesis. We made this clear in the first paragraph of Discussion “Although a direct movement of endogenous eNAD(P) is difficult to demonstrate, our data obtained using the Dex:FIN4/fin4-3 transgenic plants in which NAD biosynthesis can be spatially controlled by Dex application provide compelling support for the idea that systemic eNAD(P) originates from two sources: movement of eNAD(P) through the apoplastic route to systemic tissues as a potential mobile signal and de novo eNAD(P) accumulation in systemic tissues triggered by other mobile signals.”

We have previously reported local accumulation of eNAD(P) in pathogen-inoculated leaves (Figures 6a and 6b in Zhang et al., 2009 Plant J 57, 302-312). The measurement was done by collecting apoplastic washing fluids (AWFs). We also measured NAD(P) leakage by taking leaf disks from pathogen-inoculated leaves, similarly to ion leakage assays (Wang et al., 2019 Nat Commun). As the reviewer suggested, we measured eNAD(P) levels in the local Psm-inoculated fin4 leaves. The results are presented in Supplementary Fig. 7a, b. Local accumulation of eNAD(P) in the fin4 mutant is dramatically reduced, indicating that pathogen-induced local accumulation of eNAD(P) is also dependent on FIN4. The sentence “the fin4-3 mutant accumulated dramatically reduced eNAD(P) levels in the local leaves inoculated with Psm

(Supplementary Fig. 7a,b)” has been added in the place where systemic eNAD(P) accumulation in *fin4* is described.

2) *In the same context: It is shown that exogenously added NHP results in local and systemic accumulation of eNAD(P) (Fig. 6a-d), and that the systemic accumulation of eNAD(P) upon local induction of biological SAR is attenuated in the NHP biosynthesis mutant fmo1. The question that arises here: if eNAD(P) accumulates locally upon bacterial inoculation, is this local accumulation FMO1-dependent? Such a finding would show that NHP induces eNAD(P) release in the local tissue under biological SAR conditions, corroborate the conclusions drawn from the exogenous NHP-data (Fig. 6a-d), and strengthen the movement-hypothesis.*

A: We thank the reviewer for this simple but very informative experiment. We measured eNAD(P) levels in the local *Psm*-inoculated *fmo1* leaves. The results are presented in Supplementary Fig. 11a, b. eNAD(P) levels in the pathogen-inoculated local leaves of *fmo1* are not significantly different from those in the wild type. This result indicates that FMO1 does not play a significant role in eNAD(P) accumulation in the pathogen-infected leaf tissues (assuming most of the cells are under directly attack by pathogens in these leaf tissues; if only some cells are attacked by pathogens, FMO1 will be important for NAD(P) release from adjacent healthy cells). The eNAD(P) in the *fmo1* local leaves should be able to move to upper systemic leaves (Fig. 6g, h). Since the *fmo1* mutant is completely defective in SAR, the eNAD(P) levels in the systemic leaves of *fmo1* are not sufficient for activating effective SAR. Therefore, the FMO1-dependent de novo eNAD(P) accumulation in the systemic leaves plays a crucial role in the establishment of SAR.

We re-wrote this part in Results as “Furthermore, while eNAD(P) levels in the local *Psm*-infected leaves of *fmo1* were not significantly different from those in the wild type (Supplementary Fig. 11a, b), eNAD(P) levels in the systemic leaves of *fmo1* during biological induction of SAR were significantly lower than those in the wild type (Fig. 6g, h). Since the *fmo1* mutant is completely SAR defective, these results indicate that NHP-mediated systemic eNAD(P) accumulation plays a crucial role in the establishment of SAR.”

We also changed the discussion part to “Interestingly, although the NHP biosynthesis mutant *fmo1* accumulates significantly reduced eNAD(P) levels in systemic leaves upon SAR induction (Fig. 6g, h), eNAD(P) levels in the inoculated local leaves are comparable in *fmo1* and the wild type (Supplementary Fig. 11a, b). Since SAR is completely absent in the *fmo1* mutant, these results suggest that eNAD(P) originating from the local leaves is not sufficient for activating effective SAR and that de novo eNAD(P) accumulation in systemic tissues triggered by NHP and possibly other FMO1-dependent mobile signals is essential for SAR establishment.”

3) *Determination of extracellular NAD(P) levels: It would be helpful to be informed in the results section directly that eNAD(P) levels were assessed by analyzing apoplastic washing fluids of leaves. Moreover, the (e)NAD(P) levels are always given as relative levels which unfortunately does not provide any information on the actual, absolute quantities of NAD(P) or eNAD(P) in the leaf tissue. The absolute levels are of great interest, however, and provide information about the physiological levels of (e)NAD(P) involved in SAR induction. This could then be directly compared with the exogenous (millimolar) amounts of NAD(P) used to induce SAR chemically (Fig. 2f).*

A: We thank the reviewer for this suggestion. A sentence “Note that eNAD(P) levels were assessed by analyzing apoplastic washing fluids (AWFs) of the indicated leaves.” has been added following the description of the first eNAD(P) measurement result in the text.

We understand that the absolute levels of eNAD(P) are of great interest. We previously estimated pathogen-induced eNAD(P) levels in the inoculated leaves by infiltrating different concentrations of NAD(P) into the apoplast of healthy leaves and simultaneously collecting AWFs from these leaves and pathogen-inoculated leaves (Figures 6a and 6b in Zhang et al., 2009 Plant J 57, 302-312). The results showed that eNAD(P) levels after *Pst* DC3000/*avrRpt2* infection were similar to those after infiltration with 0.4 mM NAD⁺ and 0.8 mM NADP⁺.

To estimate eNAD(P) levels in the systemic leaves, we infiltrated three lower leaves with 1 mM MgCl₂, *Psm*, water, 0.5 mM NAD⁺, or 1 mM NAD⁺. Upper systemic leaves were collected and weighed for AWF collection. For MgCl₂ and *Psm*, the systemic leaves were collected at 48 hours after the infiltration, and for water, 0.5 mM NAD⁺, and 1 mM NAD⁺, the systemic leaves were collected at 4 hours after the infiltration. After measurement, the eNAD levels were calculated to μg/g fresh weight (FW) based on a standard curve. The eNAD levels in the systemic leaves at 48 hours after local inoculation with *Psm* is in the range of 30-40 μg/g FW, which is similar to those in the systemic leaves at 4 hours after local infiltration of 0.5 mM NAD⁺.

We added this result in the text as “To compare systemic eNAD(P) accumulation during SAR induction and exogenous NAD(P)⁺ treatment, we infiltrated lower leaves with 1 mM MgCl₂, *Psm*, water, 0.5 mM NAD⁺, or 1 mM NAD⁺. Systemic leaves were collected four hours after the water and NAD⁺ treatments and 48 hours after the MgCl₂ and *Psm* treatments for eNAD assays. The systemic eNAD levels upon SAR induction was in the range of 30-40 μg/g fresh weight (FW), which is comparable to those after 0.5 mM NAD⁺ treatment (Supplementary Fig. 6).” We hope that this result could resolve the reviewer’s concern.

4) The SAR resistance assays with *lecrk-IV* and *bak1/bkk1* mutants in the current study and the previous, Wang et al. (2019) study indicate attenuated SAR in these lines; however, a small SAR response seems to be present in these lines robustly (including a residual transcriptional SAR response in *lecrk*). This indicates the existence of another downstream signaling component independent of these genes. The same is seen for the NHP-triggered SAR (Fig. 7): a residual, NHP-triggered but *lecrk-VI.2/bak1/bkk1*-independent pathway to SAR. This should be discussed and illustrated in the final model (see below). Even in *fin4*, a small NHP-response seems to exist (Fig. 4). Could you explain this? Might there be a small eNAD(P)-independent pathway to SAR?

A: We thank the reviewer for this interesting question. We have added in Discussion “Since none of the mutants used in this study, including *rbohD*, *rbohF*, *fin4-3*, *lecrk-VI.2*, and *bak1-5 bkk1*, can completely block NHP-induced systemic immunity (Fig. 7c and Fig. 8i), a modest NHP-dependent but ROS- and eNAD(P)-LecRK-VI.2/BAK1-independent pathway to SAR seems to exist.” to reflect these results. We also modified the model to be consistent with these results. For the small NHP response in *fin4*, it could be due to a small eNAD(P)-independent pathway to SAR, but it could also be due to the residual activity of the *fin4* mutant protein, as the *fin4-3* mutant is not a null mutant (null mutant is not viable). We also know that redundancy exists in the *RBOH*, *LecRK*, and *SERK* gene families and we are making higher-order mutants of LecRKs to test the redundancy.

5) Regarding *rboh*/ROS-dependency of NHP-induced SAR and eNAD release: The authors state that “NHP induces systemic eNAD(P) accumulation and systemic immunity through RBOH-produced ROS”. On the basis of the presented data, this is an overinterpretation. I agree that the NHP-induced responses are attenuated in the *rboh* lines, but there are still considerable NHP-inducible resistance- and eNAD(P)-responses in these lines (Fig. 8). So, the statements in the manuscript throughout (including abstract and final model) on the key role of ROS should be toned down to some extent (e.g. “Rboh-dependent ROS formation positively affects but is not fully required for the NHP-triggered NAD(P) leakage and SAR responses”).

A: We agree with the reviewer. We have made changes in the abstract, the model, and elsewhere in the manuscript to indicate that NHP-induced signaling is largely but not fully depends on RBOH/ROS.

6) The ANOVA of the treatment-related (SAR, NHP) resistance data tests for differences in the induction of resistance in different lines but unfortunately does not test whether differences between treatment samples and control samples exist within individual lines (e.g. Fig. 1d,f; Fig. 7a). However, I think that this information is of interest (see my comments 4 and 5). Therefore, I suggest to compare for both the treatment-related and the genotypic differences in all of these data sets by ANOVA (as performed for eNADP levels in Fig. 8h, for example).

A: As the reviewer suggested, two-way ANOVA analyses in Fig. 1d,f; Fig. 7b,c; Fig. 8i have been changed to one-way ANOVA analyses. We kept two-way ANOVA analysis for Fig. 7a, since one-way ANOVA cannot reveal the effects of genotypes on the treatment in this case. The differences between treatment and control in Fig. 1d,f; Fig. 7b,c; Fig. 8i are discussed in the manuscript.

7) In my opinion, the final model Fig. 8k) should be improved by focussing on the actually, experimentally obtained evidence about eNADP, NHP and ROS in this and the Wang et al. (2019) study: A) in the local, 1° tissue, an arrow from NHP to eNAD(P) is missing (see results on NHP-induced local eNADP accumulation, Fig. 6a,b). Otherwise, an NHP-independent but eNAD-dependent signaling pathway is implicated, which is not the case (NHP accumulation is indispensable for SAR; NHP-deficient *ald1* and *fmo1* mutants are fully compromised in SAR). B) The depicted roles of G3P, AzA, NO, and volatiles in NHP-induced SAR are hypothetical and not experimentally verified, so I suggest to remove them from the model. C) As stated above, the function of ROS seems over-stated because the model indicates incorrectly that all the NHP-triggered SAR signaling would proceed via ROS. D) In my opinion, SA and NPR1, two key SAR components, are missing in the model. It was shown in the Wang et al. (2019)-study that eNADP-triggered SAR requires NPR1 and strongly depends on *sid2*, so SA and NPR1 act downstream of eNAD(P) [and also downstream of NHP – see Hartmann et al., Cell 2018; Liu et al., Plant Cell 2020; Yildiz et al. Plant Physiol 2021). Thus, it would make sense to place SA and NPR1 downstream of eNAD(P) in the model. E) A modest NHP-dependent but *lecrk-VI.2/bak1/bkk1*-independent pathway to SAR seems to exist (see comment 4).

A: We agree with the reviewer and have modified the model based on the results of this manuscript. We have made the following changes: (1) a thin arrow is added between 1° NHP to

1° eNAD(P) to indicate that NHP can induce eNAD(P) accumulation at the infiltrated leaf tissues and a thick arrow is added from “PTI, DTI, ETI” to 1° eNAD(P) to indicate that eNAD(P) still accumulates in pathogen-inoculated leaf tissues of the *fmo1* mutant; (2) G3P, AzA, SA, NO, Volatiles are removed from the model since they are not tested in the manuscript; (3) a question mark (?) is added on the thin arrow from 1° eNAD(P) to LecRK-VI.2/BAK1 to indicate that the contribution of this part is uncertain, which is to reflect the fact that even though 1° eNAD(P) may move to systemic tissues in *fmo1*, this mutant is fully defective in SAR; and (4) a thin arrow is added between 1° NHP and SA/NPR1 to indicate that a modest NHP/FMO1-dependent but ROS- and eNAD(P)-LecRK-VI.2/BAK1-independent pathway may exist. Since the *fin4* mutant is not a null mutant and redundancy exists among *RBOH* genes and *LecRK* genes, it is still unclear whether NHP signaling fully depends on ROS and eNAD(P)-LecRKs. Furthermore, the components in the potential ROS- and eNAD(P)-LecRK-VI.2/BAK1-independent pathway are unknown. Thus, a question mark (?) is added on this arrow to reflect these uncertainties. We hope that the changes made in the model have satisfactorily addressed the reviewer’s points.

8) *Discussion: the above-mentioned downstream function of SA and NPR1 in NHP- and eNAD(P)-mediated SAR are not properly discussed in my opinion. Also, it is worth considering that TGA transcription factors function as key downstream components of the NHP-triggered (transcriptional) SAR response (Yildiz et al., PCE 2023).*

A: We have added SA/NPR1 in the model and briefly mentioned SA and NPR1 in the discussion (but did not talk much about them). The added part is in the sentence “The new eNAD(P) further activates the receptor complex to boost the strength of SAR signaling, triggering the downstream SA/NPR1-dependent immune responses.” Since the manuscript does not have results directly related to SA and NPR1 (except in Fig. 1d, f where *npr1-3* was used as a control), we focused the discussion on NHP, ROS, eNAD(P) and their relationships. These relationships are complex and unclear, whereas SA and NPR1 have been studied extensively. In addition, the discussion part is already quite long, we are afraid that adding more information about SA and NPR1 will dilute the main idea of the manuscript.

We thank the reviewer for the suggestion of considering TGA transcription factors. Since TGAs function together with NPR1 downstream of NHP, we expect that they are also required for eNAD(P) signaling. We will test the *tga* mutants as the reviewer suggested. We cited Hartmann et al., Cell 2018; Liu et al., Plant Cell 2020; Yildiz et al. Plant Physiol 2021 in the discussion. We will cite Yildiz et al., PCE 2023 in reports about TGAs in the future.

9) *Introduction, line 39: “Plants lack ... adaptive immunity ...”: Doesn’t SAR represent a form of adaptive immunity?*

A: Adaptive immunity is one of the two immune strategies in animals (the other being innate immunity). It has unique components such as antibodies, memory B cells, and memory T cells that result from somatic hypermutations and genetic recombination. Plants lack such components and are believed to only have innate immunity. It is known that somatic mutations (also genetic recombination, epigenetic changes, etc.) may occur during SAR, but these mutations are random and unpredictable. Currently we may say that SAR is analogous to adaptive immunity, but not a form of adaptive immunity. This might change in the future.

REVIEWERS' COMMENTS

Reviewer #1 (Remarks to the Author):

The authors have clearly responded to all my comments and questions.

Reviewer #2 (Remarks to the Author):

The authors have carefully addressed all of the issues I had raised for the first manuscript version, including the suggestions for experimental improvements. This study provides interesting new insights into the molecular events underlying SAR in plants, and I thus recommend acceptance of the revised manuscript.